



# Measurement Report: Bio-physicochemistry of tropical clouds at Maïdo (Réunion Island, Indian Ocean): overview of results from the BIO-MAÏDO campaign

Maud Leriche[1,2], Pierre Tulet[3], Laurent Deguillaume[1,4], Frédéric Burnet[5], Aurélie Colomb[1], Agnès Borbon[1], Corinne Jambert[3], Valentin Duflot[6], Stéphan Houdier[7], Jean-Luc Jaffrezo[7], Mickaël Vaïtilingom[8], Pamela Dominutti[1,7], Manon Rocco[1,#], Camille Mouchel-Vallon[3], Samira El Gdachi[3,6], Maxence Brissy[1,9], Maroua Fathalli[5], Nicolas Maury[5], Bert Verreyken[10,11,6,*], Crist Amelynck[10,11], Niels Schoon[10], Valérie Gros[12], Jean-Marc Pichon[4], Mickael Ribeiro[1], Eric Pique[3], Emmanuel Leclerc[3], Thierry Bourrianne[5], Axel Roy[5], Eric Moulin[5], Joël Barrie[5], Jean-Marc Metzger[13], Guillaume Péris[14], Christian Guadagno[14], Chatrapatty Bhugwant[14], Jean-Mathieu Tibere[14], Arnaud Tournigand[14], Evelyn Freney[1], Karine Sellegri[1], Anne-Marie Delort[9], Pierre Amato[9], Muriel Joly[9], Jean-Luc Baray[1,4], Pascal Renard[1], Angelica Bianco[1], Anne Réchou[6], Guillaume Payen[13]

[1]Laboratoire de Météorologie Physique (LaMP), UMR 6016, CNRS, Université Clermont Auvergne, Aubière, 63178, France
[2]Centre pour l'étude et la simulation du climat à l'échelle régionale, Département des sciences de la terre et de l'atmosphère (ESCER), Université du Québec à Montréal, Montréal, H2X 3Y7, Canada
[3]Laboratoire d'Aérologie (LAERO), UMR 5560, CNRS, Université Paul Sabatier, IRD, Toulouse, 31400, France
[4]Observatoire de Physique du Globe de Clermont-Ferrand (OPGC), UAR 833, CNRS, Université Clermont Auvergne, Aubière, 63178, France
[5]Centre National de Recherches Météorologiques (CNRM), UMR 3589, CNRS, Université de Toulouse, Météo-France, Toulouse, 31057, France
[6]Laboratoire de l'Atmosphère et des Cyclones (LACy), UMR 8105, CNRS, Université de la Réunion, Météo-France, Saint-Denis de la Réunion, 97744, France
[7]Institut des Géosciences de l'Environnement (IGE), UMR 5001, CNRS, IRD, Université Grenoble Alpes, Grenoble, 38000, France
[8]Laboratoire de Recherche en Géosciences et Énergies (LaRGE), EA 4539, Université des Antilles, Pointre-à-Pitre, 97110, France
[9]Institut de Chimie de Clermont-Ferrand (ICCF), UMR 6296, CNRS, Université Clermont Auvergne, Aubière, 63178, France
[10]Royal Belgian Institute for Space Aeronomy (BIRA-IASB), Brussels, B-1180, Belgium
[11]Departement of Chemistry, Ghent University, Ghent, B-9000, Belgium
[12]Laboratoire des Sciences du Climat et de l'Environnement (LSCE), UMR 8212, CNRS, CEA, Université Versailles Saint Quentin, Gif-sur-Yvette, 91198, France
[13]Observatoire des Sciences de l'Univers de La Réunion (OSUR), UAR 3365, Saint-Denis de la Réunion, 97744, France
[14]ATMO-Réunion, Sainte-Marie, 97438, France
*Now at the Royal Belgian Institute for Space Aeronomy (BIRA-IASB), Brussels, B-1180, Belgium, and at Gembloux Agro-Biotech, University of Liège, Gembloux, B-5030, Belgium
#Now at Instituto de astronomia, geofísica e ciências atmosfericas (IAG), Universidade de São Paulo (USP), Rua do Matão, 1226, Butantã, São Paulo, SP – 05508-090, Brazil

*Correspondence to*: Maud Leriche (m.leriche@opgc.fr) and Pierre Tulet (pierre.tulet@aero.obs-mip.fr)



**Abstract.**

The BIO-MAÏDO (Bio-physicochemistry of tropical clouds at Maïdo (Réunion Island): processes and impacts on secondary organic aerosols formation) campaign was conducted from the 13th of March to the 4th of April 2019 on the tropical Réunion Island and implied several scientific teams and state-of-the-art instrumentation. The campaign was part of the BIO-MAÏDO

project with the main objective is to improve our understanding of cloud impacts on the formation of secondary organic aerosols (SOA) from biogenic volatile organic compounds (BVOC) precursors in a tropical environment. Instruments were deployed at five sites: a receptor site, the Maïdo observatory (MO) at 2165 m asl, and four sites along the slope of the Maïdo mountain. The obtained dataset includes measurements of the gas-phase mixing ratio of volatile organic compounds (VOC), the characterization of the physical, chemical, and biological (bacterial diversity) properties of aerosols and the characterization

of the physical, chemical and biological (identification of viable bacteria through culture-based approaches) properties of the cloud water. In addition, the turbulent parameters of the boundary layer, radiative fluxes, and emissions fluxes of BVOC from the surrounding vegetation were measured to help with the interpretation of the observed chemical concentrations in the different phases. Dynamical analyses show two preferred trajectories routes for air masses arriving at MO during the daytime both corresponding to the return branches of the trade winds associated with the up-slopes thermal breezes. These air masses

likely encountered cloud processing during transport along the slope. The highest mixing ratio of oxygenated VOC (OVOC) were measured above the site located in the endemic forest and the highest contribution of OVOC to total VOC at MO. Chemical composition of particles during the daytime shows a higher concentration of oxalic acid and a more oxidized organic aerosol at MO than at other sites along the slope. This is a signature of photochemical aerosols aging along the slope potentially influenced by cloud processing. Despite an in-depth analysis of organic compounds in cloud water, around 80% on average of

dissolved organic compounds is undefined highlighting the complexity of the cloud organic matter.

## 1. Introduction

Aerosols are essential components in the atmosphere as a result of their role in the radiative budget of the earth, including their indirect impact by acting as cloud condensation nuclei (CCN) and ice nuclei (IN) in the formation of cloud droplets and ice crystals. Their impact on climate is still uncertain (Boucher et al., 2013). Aerosols are also a major contributor to air pollution

and their health effects have been demonstrated (World Health Organization, 2021). However, there are still major uncertainties in the formation and transformation of atmospheric aerosols. These uncertainties need to be lifted to understand the impacts of these particles on air quality, health, and climate change. Atmospheric aerosols have a complex chemical composition and the organic fraction, which contributes significantly to the total mass of fine particles (Jimenez et al., 2009), is still the least characterized to date. 3D atmospheric chemistry models are globally unable to reproduce the observed amount,

oxidation level, and spatial distribution of organic aerosols (Heald et al., 2011; Jathar et al., 2016; Pai et al., 2020). Among this organic fraction, a major part of the mass is of secondary origin (Zhang et al., 2007). The main precursors of the secondary organic aerosols (SOA) are natural compounds (isoprene and terpenes) and aromatics from anthropogenic origin. Even if the





chemical reactivity in the gaseous phase of these volatile organic compounds (VOC) is relatively well known, the nature and the potential of SOA formation of their oxidation products are still uncertain. Biogenic VOC (BVOC) from terrestrial vegetations are particularly important since they dominate the global emission of nonmethane hydrocarbons in the atmosphere (Guenther et al., 2012). The oxidation of BVOC in the atmosphere forms less volatile oxidized chemical species, which participate in SOA formation through various complex processes (Shrivastava et al., 2017). These oxidized products are soluble in water where they are photo-oxidized (Ervens et al., 2011). The chemical reactivity in aqueous phase is different than in the gas phase and can lead to the formation of low volatility compounds (Carlton et al., 2007; Liu et al., 2009) including oligomers (Renard et al., 2015). It is now well established that the aqueous phase oxidation contribution to the SOA formation is significant (McNeil, 2015; Su et al., 2020 in polluted conditions) but still misunderstood in term of processes and badly represented in 3D models (Ervens, 2015). The main contributors to SOA formation from cloud chemistry are known to be low volatility organic acids coming mainly from the photo-oxidation of glyoxal and methylglyoxal (Ervens et al., 2011). Recently, Tsui et al. (2019) showed that isoprene epoxydiols (IEPOX) could be a significant contributor to SOA formation from cloud chemistry. The presence of bacteria in cloud water also has a potential impact on cloud chemical composition (Vaïtilingom et al., 2013; Khaled et al., 2021). A recent study on at the global scale estimates that microbial processes might lead to a loss of water-soluble organic content in cloud droplets of the same order of magnitude than the loss from chemical processes (Ervens and Amato, 2020). However, this estimation is very uncertain, and, to the best of our knowledge, no study has assessed the effect of microbial processes in cloud droplets on SOA formation.

Humid tropical atmospheres, which are characterized by high biogenic emissions and a high occurrence of fogs and clouds, is particularly favorable to SOA formation through biogenic precursors via cloud multiphase chemistry. Réunion Island is a small tropical island in the Indian Ocean at the east of Madagascar Island. Anthropogenic sources are limited in the island and are mainly in the coastline area, as the island is far from the impact of large anthropogenic emission sources (Duflot et al., 2019). La Réunion is a volcanic island with an abrupt topography and high mountainous area (Piton des Neiges, 3070 m) and presents 100 000 ha of native ecosystems (Duflot et al., 2019). Lesouëf et al. (2011; 2013) described the complex atmospheric dynamic on the island, which is, at the large scale, affected by easterly/south-easterly trade winds near the ground and westerlies in the free troposphere. Due to the strengthening of the large-scale subtropical subsidence at night, air masses at high altitude are disconnected from local and regional anthropogenic sources during the night and early morning.

The Maïdo atmospheric observatory (altitude 2165 m) (Baray et al., 2013), close to the Piton Maïdo (2190 m), is in the northwest part of the island and offers a unique opportunity to study SOA formation processes in the humid tropical atmosphere. The slope of the Maïdo, west of the observatory, is covered with tropical forests characterized by endemic tree species *Acacia heterophylla (Fabaceae)*, plantations of the coniferous species *Cryptomeria japonica (Taxodiacae)* and the *Acacia heterophylla* forest, locally called "Tamarinaie" (Duflot et al., 2019). A first campaign devoted to cloud-aerosols interaction (Duflot et al., 2019) in March-April 2015 showed the potential of the observatory to study the formation of SOA influenced by clouds with the diurnal formation of clouds on the slope below the observatory over Tamarinaie, which emits



isoprene and terpenes, and their dissipation at the level of the observatory. Measurements performed during this campaign showed high levels of formaldehyde, a product of isoprene oxidation and the presence of viable bacteria in cloud water.

The BIO-MAÏDO (Bio-physicochemistry of tropical clouds at Maïdo (Réunion Island): processes and impacts on secondary organic aerosols formation) project was designed in this context with three main objectives: (i) to understand which are the

main formation pathways of SOA in humid tropical atmosphere (gaseous phase versus aqueous phase); (ii) to improve multiphase processes leading to SOA formation in 3D model; (iii) to examine whether the presence of bacteria in aqueous phase could contribute to SOA formation. The strategy of BIO-MAÏDO is based on an intensive field campaign using the Maïdo observatory facilities, in synergy with modelling studies using: two lagrangian particle dispersion models FLEXPART-AROME (Verreyken et al., 2019) and Meso-CAT (Rocco et al., 2022), a 0D cloud chemistry model (CLEPS, Mouchel-Vallon

et al., 2017; Rose et al., 2018) and a 3D model coupling meteorology and chemistry and including a cloud chemistry module (Meso-NH, Lac et al., 2018; http://mesonh.aero.obs-mip.fr/, last access: 20 June 2023).

The aim of the present paper is to present an overview of the results obtained from the campaign. The general strategy of the campaign and the description of the five sampling sites are provided. Then, main results obtained from measurements are summarized describing the boundary layer evolution and cloud cycles, gas, aerosol and cloud chemistry.

**2. Strategy of the campaign and sites description**

**2.1 Strategy**

In the general context of the three main objectives of the BIO-MÄIDO project presented above, the field campaign aims to: (1) characterize the chemical and biological composition of air and cloud samples and identify main sources of gases and aerosols; (2) characterize the dynamics and the evolution of the boundary layer and the macro and micro-physical properties

of clouds; (3) determine case studies for modelling work with CLEPS and Meso-NH.

The campaign took place from the 13th of March to the 4th of April 2019. This period was chosen to include frequent periods of formation of low-level cloud and of convective precipitating clouds along the slopes of the Maïdo (Duflot et al., 2019), a high UV index (12) and high temperatures (end of the southern summer). Moreover, the campaign took place during the 2-years observation period of the OCTAVE project (Oxygenated Compounds in the Tropical Atmosphere: Variability and

Exchanges, http://octave.aeronomie.be, last access: 20 June 2023) during which complementary instrumentations were deployed at the Maïdo observatory to characterize oxygenated volatile organic compounds (OVOC) mixing ratios. The main objective of OCTAVE is to improve the climatology of the global budget of OVOC and their role in tropical regions (Rocco et al., 2020; Simu et al., 2021; Verreyken et al., 2021).

The field campaign included five sampling sites (Figure 1). Except for the Maïdo observatory, these sites are all located along

the northwestern slope to the Maïdo site, identified as one of the two main paths for dominant winds. At the mid-morning almost every day, clouds form on the slope of the Maïdo, and in general evaporate at the altitude of the observatory (Duflot et al., 2019).



An innovative instrumentation was deployed, including for instance: three proton-transfer-reaction mass spectrometer (PTR-MS) for online analysis of VOC, one of which was operated at high frequency and coupled to an ultrasonic anemometer
allowed measurements of BVOC fluxes; a tethered balloon to capture microphysical characteristics of clouds; an aerosol chemical speciation monitor (ACSM), which provides online measurements of the non-refractory submicron chemical composition (NR-PM$_1$); a new generation of cloud droplet impactor to accumulate cloud water and allow biological (bacteria diversity and number concentration, ATP quantification) and detailed chemical analyses further in the lab. For instance, dissolved organic compounds have been intensively investigated in cloud water to characterize their atmospheric sources,
evaluate the chemical and biological processes occurring in the air during the transport of the organic matter, and to assess their partitioning among the gas and aqueous phases.

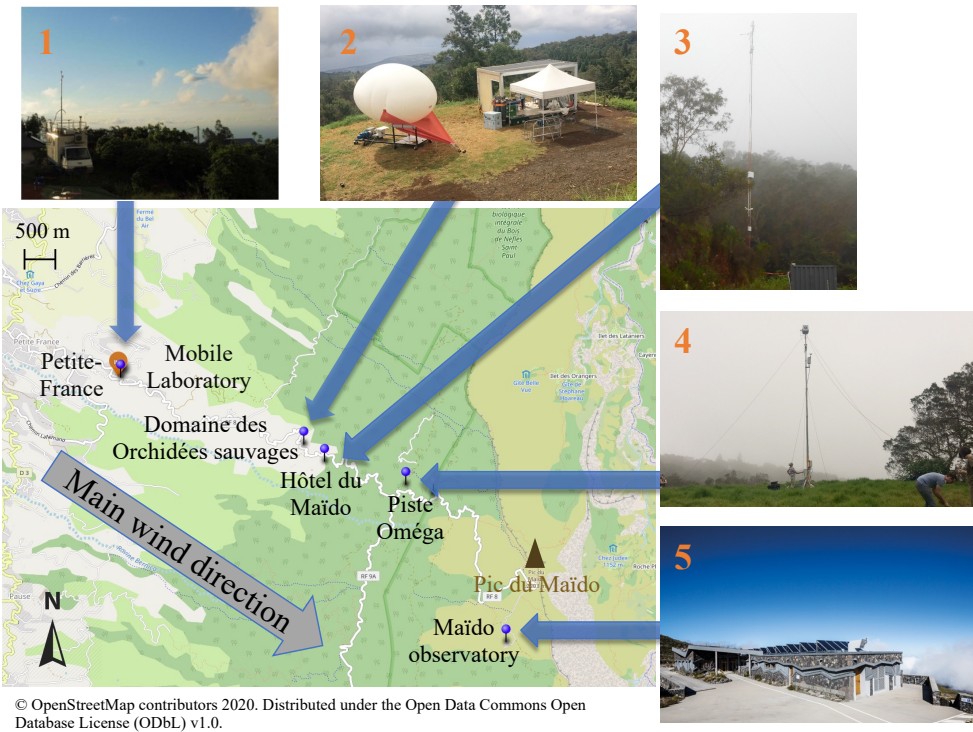

**Figure 1. Location of the five-instrumented sites during the BIO-MAÏDO campaign.**

During the whole campaign, FLEXPART (Pisso et al., 2019) coupled with the AROME operational forecasts at 2.5 km of
horizontal resolution (Verreyken et al., 2019) was used to analyze the regional origin (marine boundary layer, free troposphere) of the air masses observed at the Maïdo area (https://geosur.osureunion.fr/public_html/cgi-bin/web/display_biomaido_v2.py, last access: 20 June 2023). This information has been supplemented by back-trajectories computed with Meso-CAT (Rocco et al., 2022) resulting from the coupling between high-resolution Meso-NH simulations and the lagrangian tool CAT (Computing Advection-interpolation of atmospheric parameters and Trajectory tool; Baray et al., 2020). These back-trajectories allowed



assessing the local contribution of biogenic, anthropogenic, and marine source area in the chemical composition of air masses sampled at the sampling sites and to determine which days present a dynamical connection between the sites.

## 2.2 Petite France (PF-1): a rural site under urban influence

Petite France (PF-1, 965 m asl, 21°02'33.3"S 55°19'32"E) is a neighborhood/district of the municipality of Saint-Paul. The land-cover around PF-1 comprises mainly of residential areas, grassland, and sugar cane plantations. A part of the instruments

is deployed inside the monitoring truck of Atmo-Réunion (https://atmo-reunion.net/, last access: 20 June 2023), the association in charge of the air quality monitoring on the island. Instrumentation deployed at PF-1 aimed at characterizing the chemical composition of the air including gases and $PM_{10}$. The instrumentation onboard the truck included analyzers for ozone, carbon monoxide, nitrogen oxides, sulfur dioxide, a particle counter for $PM_{2.5}$ and a proton-transfer-reaction quadrupole mass spectrometer (PTR-QMS) for online VOC characterization. The chemical and biological composition of $PM_{10}$ was analyzed

from pure quartz fiber filters sampled twice a day during night and day with a high-volume sampler. Various chemical analyzes on filters were performed in the lab to quantify the major chemical constituents and specific chemical tracers. The carbonaceous fraction of particles (EC and OC) was analyzed with a Sunset Lab analyzer (using the EUSAAR2 thermo-optical protocol, Cavalli et al., 2010). The major ions components were measured by ion chromatography using an ICS300 Chromatograph (dual-channel, Thermos-Fisher) following the standard protocol described in Jaffrezo et al. (2005). Anhydro-sugars and

saccharides were analyzed by high-performance liquid chromatography with pulsed amperometric detection (HPLC-PAD, using an ICS 5000+ Chromatograph, Samake et al., 2019). The analysis of organic acids was conducted using a HPLC-MS (GP40 Dionex), with negative mode electrospray ionization (Borlaza et al., 2021). The diversity of bacteria in $PM_{10}$ samples was investigated by high-throughput sequencing (Illumina) of metabarcoded ribosomal gene amplicons, from whole genomic DNA extracted using the commercial DNeasy PowerWater kit (Qiagen). Polymerase chain reaction (PCR) amplification was

performed using the primers 515F and 806R, as recommended by the Earth Microbiome Project (Caporaso et al., 2012). The sequence data obtained from Illumina MiSeq (2x250bp) were analyzed through the FROGS pipeline (Escudié et al., 2018) using Silva 132 as the reference database (Quast et al., 2013), as in Péguilhan et al. (2021). In addition, a ceilometer was deployed to characterize the boundary layer evolution and the cloud cycle. Table 1 summarizes the instruments deployed at PF-1 and the associated measured parameters.




Table 1. Instrumentations deployed at PF-1 and associated measured parameters.

| Instrument | Measured parameter | Sampling frequency | Institution in charge |
|---|---|---|---|
| Ultrasonic wind sensor Windsonic , Gill Instruments | Wind | 15 min | Atmo-Réunion |
| Humidity and temperature sensor EE210, E+E Elektronik GmbH | Temperature Relative humidity | 15 min | Atmo-Réunion |
| CO analyzer T300, Teledyne API | Mixing ratio of CO | 15 min | Atmo-Réunion |
| Ozone analyzer O342M, Environnement SA | Mixing ratio of $O_3$ | 15 min | Atmo-Réunion |
| NO/NO2/NOx analyzer T200, Teledyne API | Mixing ratio of NO, $NO_2$, NOx, | 15 min | Atmo-Réunion |
| SO2 analyzer 43i, Thermo Fisher Scientific Inc. | Mixing ratio of $SO_2$ | 15 min | Atmo-Réunion |
| PTR-QMS, Ionicon Analytik GmbH | Mixing ratio of COV | 1 min | LSCE |
| Aerolaser AL4021, Aero-Laser GmbH | Mixing ratio of HCHO | 1 min 28/03-04/04 | LaMP |
| Condensation Particle Counter MAGIC CPC, Aerosol Devices Inc. | Number concentration of particles with diameter from 5 nm to 2.5 μm | 10 s | Atmo-Réunion |
| High Volume Sampler (Hi-VOL) Digitel DA80, Megatec – filter | $PM_{10}$ mass chemical concentration and bacterial diversity | Day and night 10-12 h | IGE |
| PQS1 radiometer, Kipp &Zonen | Photosynthetic Active Radiation (PAR) | 15 min | CNRM |
| Ceilometer CT25K, Vaissala | Cloud base height Backscatter profile | 1 min 1 min | LACy |



**2.3 Domaine des Orchidées Sauvages (DOS-2): a strategic site to observe cloud cycle**

Domaine des Orchidées Sauvages (DOS-2, 1465 m asl, 21°03'07"S 55°21'11"E) is a large private property where the tethered
balloon was operated. Moreover, several sets of devices were also deployed there: a meteorological station, complementary
probes monitoring the size spectrum of particles, a present weather visibility sensor associated with a droplet size spectrometer
to characterize the microphysical properties of clouds, as well as a ceilometer and the Lidar MARLEY (Mobile AeRosol
Raman Lidar for troposphEre surveY) to characterize the boundary layer evolution, the cloud cycle, and the vertical profile of
aerosols. The land cover around DOS-2 is composed mainly of a mix of grassland and forests.

The tethered balloon was equipped with an ultrasonic anemometer and a temperature probe at high frequency to estimate the
heat and momentum fluxes and the turbulent kinetic energy by eddy covariance. At the beginning of the morning, the tethered
balloon was operated in clear sky with an aerosol probe whereas it was operated with cloud sensors at the end of the morning
when cloud appeared. The tethered balloon was operated for 21 days for 144 hours of measurements. Following Fathalli et al.
(2022), the adopted strategy was alternating vertical soundings and levels at constant altitude (20 minutes for turbulence, or 5
to 10 minutes for cloud microphysics statistical representativeness) for each flight. Table 2 summarizes the instruments
deployed at DOS-2 and the associated measured parameters.





Table 2. Instrumentations deployed at DOS-2 and associated measured parameters.

| Instrument | Measured parameter | Sampling frequency | Institution in charge |
|---|---|---|---|
| **Condensation Particle Counter CPC3788, TSI** | Total number concentration of aerosols with diameter from 2.5 nm to 2.5 µm | 1 s | CNRM |
| **Scanning Mobility Particle Sizer SMP3080, TSI** | Size spectrum of aerosols with diameter from 10 nm to 500 nm | 3 min | CNRM |
| **Optical Particle Sizer OPC3330, TSI** | Size spectrum of aerosols with diameter from 0.3 µm to 10 µm | 5 min | CNRM |
| **CCNC Droplet Measurement Technologies** | Number concentration of CCN at S = 0.1%, 0.2% and 0.3% | 5 min for each supersaturation | CNRM |
| **Fog monitor, Droplet Measurement Technologies** | Size spectrum of droplets with diameter from 2 µm to 50 µm | 1 s | CNRM |
| **Present Weather Detector PWD22, Vaisala** | Visibility | 15 s | CNRM |
| | Rain | 15 s | |
| | Luminance | 15 s | |
| **Lidar MARLEY** | Backscatter profile | 1 min | LACy |
| **Ceilometer CS135, Campbell Scientific** | Cloud base height | 10 s | LACy |
| | Backscatter profile | 10 s | |
| **Tethered Balloon** | | | |
| **Optical particle counter (OPC), MetOne** | Size spectrum of aerosols with diameter from 0.5 µm to 10 µm | 6 s | CNRM |
| **Cloud drop probe (CDP), Droplet Measurement Technologies** | Size spectrum of droplets with diameter from 2 µm to 50 µm | 1 s | CNRM |
| **Sonic anemometer** | Wind | 20 Hz | CNRM |
| | Temperature | 20 Hz | |
| | Relative humidity | 20 Hz | |





### 2.4 Hôtel du Maïdo (HM-3): a forest area dedicated to fluxes measurements

Hôtel du Maïdo (HM-3, 1500 m asl, 21°03'16.4"S 55°21'21.4"E) is a former holiday camp located in the middle of the forest. This site was dedicated to measurements of VOC fluxes. A 24 m instrumented mast and a container had been installed on the site. Several devices were deployed on the top of the mast: an ultrasonic anemometer including a temperature probe and an analyzer of carbon dioxide and water vapor. An inlet connected to a pump inside the container had been also installed at the top of the mast. This inlet brought air inside the container to several devices: a second analyzer of carbon dioxide and water

vapor, an ozone analyzer, a proton-transfer-reaction time of flight mass spectrometer (PTR-TOFMS) for VOC measurements and an active sampling on sorbent cartridges. The comparison of measurements from both analyzers of carbon dioxide and water vapor allowed estimating the effects of the inlet on other chemical compounds measurements inside the container. Measurements of the mixing ratio of biogenic organic compounds (isoprene and monoterpenes) by PTR-TOFMS at 5 Hz were used to estimate their emissions thanks to the ultrasonic anemometer by eddy covariance. Finally, measurements of shortwave

and longwave, upward and downward radiative fluxes, as well as of visibility and photosynthetically active radiation (PAR) were operated at the bottom part of the mast. Table 3 summarizes the instruments deployed at HM-3 and the associated measured parameters.




Table 3. Instrumentations deployed at HM-3 and associated measured parameters.

| Instrument | Measured parameter | Sampling frequency | Institution in charge |
|---|---|---|---|
| **3D Sonic anemometer CSAT 3, Campbell Scientific** | Wind, temperature, relative humidity | 10 Hz | LAERO |
| **LI-7500 open path gas analyzer, LI-COR** | Mixing ratio of $CO_2$ and $H_2O$ (mast) | 10 Hz | LAERO |
| **CNR4 radiometer, Kipp & Zonen** | Up and down, longwave and shortwave radiations | 1 min | LAERO |
| **Present Weather Detector PWD22, Vaisala** | Visibility | 15 s | CNRM |
| | Rain | 15 s | |
| | Luminance | 15 s | |
| **PQS1 radiometer, Kipp &Zonen** | Photosynthetic Active Radiation (PAR) | 1 min | LAERO |
| **LI-6262 closed path gas analyzer, LI-COR** | Mixing ratio of $CO_2$ and $H_2O$ (container) | 10 Hz | LAERO |
| **$O_3$ analyzer TEI49i, Thermo Fisher Scientific Inc.** | Mixing ratio of $O_3$ | 5 min | LAERO |
| **PTR-ToF-MS 1000 ultra, IONICON** | Mixing ratio of VOC | 5 Hz | LAERO |
| **Smart Automatic Sampling System (SASS)** | Mixing ratio of VOC | 5 cartridges/day 18/03-05/04 | LaMP |


## 2.5 Piste Omega (PO-4): a forest site to sample cloud water

Piste Oméga (PO-4, 1760 m asl, 21°03'26.8"S 55°22'05.0"E) is a forest trail from the Maïdo road. The site is surrounded by forest. A 10 m mobile mast was used to install a cloud impactor (cf. Fig. 1). This collector was facing the slope from which the cloud came. A modified cloud drop probe (CDP) had been fixed on the mast just under the cloud collector to monitor the
cloud microphysical properties. It measures the droplet size distribution from 2 to 50 µm in diameter, enabling the calculation of the liquid water content (LWC) and of the effective diameter ($D_{eff}$). A meteorological station had also been fixed on the mast (T, RH, wind speed). Finally, the AEROVOCC sampler has been installed on the side of the cloud collector. AEROVOCC



was developed to sample VOC and OVOC in cloudy air. It consists of three sorbent cartridges connected to three automated pumps to control samples at a constant flow.

During the whole campaign, 14 cloud water samples have been collected (named from R1 to R14 hereafter). The mean volume of samplers was 111 mL. The sampled cloud water was further analyzed in lab for (1) pH; (2) main inorganic ions by ionic chromatography; (3) total organic carbon (TOC) by TOC analyzer; (4) targeted organic compounds by high-performance liquid chromatography (HPLC) – mass spectrometry (MS) for carboxylic acids, by HPLC with fluorescence detection (HPLC-Fluo) for carbonyl compounds, by HPLC-high resolution mass spectrometry (LC-HRMS) for amino acids, by HPLC with pulsed

amperometric detector (HPLC-PAD) for sugars, by stir bar sorptive extraction (SBSE) coupled to gas chromatography-mass spectrometry (GC-MS) for low soluble VOC; (5) hydrogen peroxide by derivatization and spectro-fluorescence and, Fe(II) and Fe(III) by complexation and ultraviolet-visible spectrophotometry (UV/Vis spectrophotometer). When enough volume of water was available, a non-target analysis was performed to investigate the complexity of the dissolved organic matter. 3 clouds were analyzed by Fourier Transform Ion Cyclotron Resonance Mass Spectrometry (FT-ICR MS), using ionization in

positive and negative polarity and multiple ionization sources to get a full picture of the composition of organic matter.

Viable culturable bacteria were investigated by culture-platting of 0.1 mL of water samples on R2A medium and incubation at 25°C in the dark. Colonies were isolated and taxonomically identified based on full length 16S rRNA gene sequences, obtained from PCR amplification using the primers 27f and 1492r, and online BLAST software available from NCBI's website, as in Vaïtilingom et al., 2012. Table 4 summarizes the instruments deployed at PO-4 and the associated measured parameters.


Table 4. Instrumentations deployed at PO-4 and associated measured parameters.

| Instrument | Measured parameter | Sampling frequency | Institution in charge |
|---|---|---|---|
| **Meteorological station** | Wind, temperature<br>Pressure, relative humidity | 1 min | CNRM |
| **Cloud drop probe (CDP), Droplet Measurement Technologies** | Size spectrum of droplets with diameter from 2 μm to 50 μm | 1 s | CNRM |
| **AEROVOCC** | Mixing ratio of VOC and OVOC | One sample by event | LaMP |
| **Cloud impactor** | Chemical composition of cloud water<br>Viable bacteria diversity | Duration of the event | LaMP |



**2.6 Maïdo observatory (MO-5): a receptor site to observe process air mass**

Maïdo observatory (MO-5, 2165 m asl, 21°04'46"S 55°22'59"E) is surrounded by mountain shrublands. Previous simulations
with the 3D Meso-NH model showed the vanishing of clouds at the same period of the year than the campaign at the
observatory altitude (Duflot et al., 2019).

The Maïdo observatory (Baray et al., 2013) can host atmospheric scientific experiments and offer the possibility for scientists
to stay on site. Several European and international observation services operate at the observatory, and associated routine
observations are done, in particular, of interest for BIO-MAÏDO: for aerosols (submicron size distribution, CCN concentration,
and total $PM_{2.5}$ number concentration), for gases (mixing ratio of ozone, nitrogen oxides, carbon monoxide and sulfur dioxide)
and basic meteorological parameters. The BIO-MAÏDO campaign also benefited from the two-year presence (October 2017
to November 2019) at the observatory of the high-sensitivity quadrupole-based PTR-MS of BIRA-IASB as part of the
OCTAVE project. This suite of instruments was complemented by the online chemical characterization of non-refractory $PM_1$
using the Time-of-Flight Aerosol chemical speciation monitor (ToF-ACSM), $PM_{10}$ filter sampling using a high-volume
sampler identical to those deployed at PF-1 (Dominutti et al. 2022b), and by the same set of instruments deployed at DOS-2
to characterize the microphysical properties of clouds. The chemical and biological analyses of $PM_{10}$ performed from filters
are the same as at PF-1. The MO-5 station is also part of the ACTRIS (Aerosol, Clouds and Trace Gases Research
Infrastructure) monitoring network and monitors aerosol size distribution and number concentration using a custom-made
differential mobility particle sizer (DMPS) with a commercially available condensation particle counter (CPC, TSI). Table 5
summarizes the instruments deployed at MO-5 and the associated measured parameters.





Table 5. Instrumentations deployed at MO-5 and associated measured parameters.

| Instrument | Measured parameter | Sampling frequency | Institution in charge |
|---|---|---|---|
| **Fourier-transform infrared spectroscope (FTIR)** | Wind, temperature Pressure, relative humidity | 1 min | OPAR/BIRA-IASB |
| **CO analyzer Horiba** | Mixing ratio of CO | 1 min | OPAR |
| **$O_3$ analyzer TEI49i, Thermo Fisher Scientific Inc.** | Mixing ratio of $O_3$ | 1 min | OPAR |
| **$NO_x$ analyzer, Environnement SA AC31M** | Mixing ratio of $NO_x$, NO, $NO_2$ | 1 min | OPAR |
| **$SO_2$ analyzer T421, Thermo Fisher Scientific Inc.** | Mixing ratio of $SO_2$ | 1 min | OPAR |
| **Condensation Particle Counter CPC3776, TSI** | Number concentration of particles with diameter from 25 nm to 1 μm | 10 s | OPAR |
| **Custom-made Differential Mobility Particle Sizer with a Condensation Particle Counter CPC3100, TSI** | Size spectrum of aerosols with diameter from 13.7 nm to 650 nm, 14 size classes | 8 min | OPAR/LaMP |
| **Aerosol Chemical Speciation Monitor ToF-ACSM, Aerodyne Research Inc.** | Chemical composition of $NR-PM_1$ | 10 min | LaMP |
| **High Volume Sampler (Hi-VOL) Digitel DA80, Megatec – filter** | $PM_{10}$ mass chemical concentration and biological composition | Day and night 10-12 h | IGE |
| **PTR-MS, Ionicon Analytik GmbH** | Mixing ratio of VOC | 2.7 min | BIRA-IASB |
| **Aerolaser AL4021, Aero-Laser GmbH** | Mixing ratio of HCHO | 1 min 13/03-27/03 | LaMP |
| **Fog monitor, Droplet Measurement Technologies** | Size spectrum of droplets with diameter from 2 μm to 50 μm | 1 s | CNRM |
| **Present Weather Detector PWD22, Vaisala** | Visibility Rain Luminance | 15 s 15 s 15 s | CNRM |



## 3. Main results

A large range of data was collected during the BIO-MAÏDO campaign. The cloud water collector deployed at PO-4 (Dominutti
et al., 2022a) allowed more efficient sampling of cloud droplets than during the FARCE campaign (Duflot et al., 2019). Table
S1 in the supplement summarizes the daily operation of all instruments deployed during the campaign. This report has been
used to identify the days with the maximum amount of information available.

### 3.1 Meteorological overview of the campaign

The meteorological environment of Reunion Island (Réchou et al., 2019), and particularly of the Maïdo area, has been
extensively studied in recent years in the frame of many measurement campaigns performed at the Maïdo observatory. Lesouëf
et al. (2011), Guilpart et al. (2017) and Foucart et al. (2018) have highlighted the main local and regional circulations that
affect measurements at the observatory. Lesouef et al. (2013) and Duflot et al. (2019) studied the evolution of the mixing
boundary layer and highlighted the superposition of several stratified layers along the Maïdo slopes. These previous results
are summarized below.

For the main meteorological circulations that affect the campaign area, Reunion Island is subject to a strong easterly/south-
easterly trade wind flow in winter and a weaker one in summer in the lowest layers of the atmosphere when the intertropical
convection zone is close to Réunion Island. The Maïdo region located to the northwest of the island is conditioned by the
convergence of two distinct flows: (i) The overflow regime due to the lifting of the trade winds over the topography of the
island. This flow mainly affects the free troposphere; (ii) The counterflow regime corresponding to the circumvention of the
trade winds around the topography. This low and medium altitude flow leads to a return flow generally located in the west and
north-west sector, downwind of the island.

More locally, thermal breezes strongly influence the weather during the daytime: near the coast, sea/land breezes are formed
and higher up, anabatic/catabatic breezes affect the slopes during the day/night. All these circulations lead to ascents on the
slopes of Maïdo almost daily with clouds formation in the middle of the morning and beginning of the afternoon, then
subsidence and stratification of the boundary layer leading to the evaporation of the clouds at the end of the daytime. The
Maïdo observatory is generally located in the trade winds overflow flow except in the middle of the day when it is located
close to a convergence zone between the overflow flow and the thermal ascents on the slopes. At night, the observatory is
located in the free troposphere.

These different mechanisms have been highlighted mainly by numerical modeling and on case studies lasting a few days. The
availability of two meteorological stations located at PF-1 (mid-slope) and MO-5 (summit) during the BIO-MAÏDO campaign,
made it possible for the first time to characterize thermal breezes on the Maido slope over a period of 27 days corresponding
to the transition period between the wet season and the dry season.

The weather conditions observed during the campaign are summarized by the wind roses resulting from local observations in
Figure 2. Fig. 2a and 2b correspond to the wind observed at PF-1 and MO-5 between 14 UTC and 04 UTC (i.e. from the end





of the afternoon to the beginning of the morning). At both stations, the wind regime is from the south and south-east, with a maximum wind speed between 2 to 3 m s$^{-1}$ at PF-1 (frequency 35%) and higher than 4 m s$^{-1}$ at MO-5 (frequency 15%). These conditions show the strong influence of the trade winds and the overflow conditions at night. The wind conditions observed during the daytime (from 04 UTC to 18 UTC; Fig. 2c and 2d) are more variable, especially at MO-5. At PF-1, the flow is essentially from the west (30 %) and of lower intensity than at MO-5 (generally between 1 and 3 m s$^{-1}$). This direction is

typical to the up flow of the air mass on the slopes by thermal breezes or by the return flow of the trade winds. At MO-5, the air masses have two opposite directions: from the north (14%) to the east (8%) and from the south (6 %) to the west (20 %). This indicates a reversal of the wind direction during the daytime, as observed by Rocco et al. (2022). These conditions were consistent with the period of transition from the wet to the dry season.

**Figure 2. Wind rose at PF-1 and MO-5 between 14 UTC and 04 UTC (nighttime) and from 04 UTC to 18 UTC (daytime) averaged**
**over the entire campaign from March 11 to April 7. The intensity is in m s$^{-1}$.**



Verreyken et al. (2020; 2021) and Rocco et al. (2020) studied the origin of the air masses measured at the Maïdo observatory by using Lagrangian trajectory tools (FLEXPART, CAT). Their results concerned the origin of air masses from the atmospheric transport at the synoptic scale and showed that the dominant large-scale air masses are easterly under the influence of trade winds and that the strongest biogenic contributions coincided with air masses passing over the northeastern part of La

Réunion. For the BIO-MAÏDO campaign, to determine the origin of the air masses arriving at PF-1 and MO at the scale of the entire island, high spatio-temporal back-trajectories have been calculated using Meso-CAT. The used Meso-NH simulations cover the whole campaign and use three embedded domains at 2 km, 500 m and 100 m of horizontal resolution. Rocco et al. (2022) first exploited these back-trajectories by combining them with soil data from the Corine 2018 land-cover database (Geoportail, https://www.geoportail.gouv.fr/, last access: 20 June 2023) to assess the origin of the air masses sampled at MO.

Moreover, the dominant circulations schemes at the scale of the highland are better highlighted by new elements provided here and obtained by calculating footprint maps using Meso-CAT in back-trajectory mode.

Figures 3 gives the footprint of PF-1 and MO-5 using back-trajectories of Meso-CAT from the 500 m of horizontal resolution domain. These footprints are computed by assembling all the back-trajectories that reached these two stations and by counting all the trajectory points per pixel of 1 km size. Two types of footprints have been calculated, a first which corresponds to the

total atmospheric column, keeping all the trajectory points (Fig. 3a and 3b), and a second for which we select only the trajectory points located less than 500 meters above the ground level and during the mid-day (between 06 UTC to 14UTC; Fig. 3c and 3d).

The total column footprint of PF-1 (Fig. 3a) clearly shows the influence of the trade winds with air masses arriving from the south-east, bypassing the island of La Reunion Island as much from the north as from the south, and arriving on PF-1 generally

with an ascent by the slopes. These air masses are well channeled and bypass the topography by the southern flank of Reunion Island. The main part of back-trajectories arrives locally from the west and is more variable and diffuse over the whole campaign period. At MO-5 (Fig 3b), there is a wider dispersion of air mass origins. The signature of the trade winds is even more visible with air masses arriving at MO-5 generally more directly and pass mainly through the south. The footprint also shows more trajectory points east of MO-5, indicating the influence of frequent ascents of air masses from the Cirque de

Mafate.

To study the mixing boundary layer air masses advected by thermal breezes, other footprints have been calculated according to the following criteria: (i) only the back-trajectories remaining in the mixing boundary layer arbitrarily set as a 500 m (above ground level) thick layer are kept; (ii) we have retained the periods of the day when there are back-trajectories coming from south-west to north-west, between 06 UTC and 14 UTC (Fig. 3c and 3d). Again, two preferred trajectories routes can be seen

for PF-1 and for MO-5, the main part of the air masses passing through the south. This means that the PF-1 measurements were able to load themselves with chemical and biological compounds over the forests located between 1000 m and 1500 m asl in the south-west of the island. The other well-marked result is the one going up the western flank of the island, whose trajectories also show a passage in the marine boundary layer. By differences with the total day integration, one can note that



most of the back-trajectories modeled at MO-5 come from the south to the north-west, which corresponds to the return branches
of the trade winds associated with the up-slopes thermal breezes.

**Figure 3. Natural logarithm of the number of back-trajectory points arriving at Petite France (left) and Maïdo (right) from 15 March to 8 April, per scare of 0.01° (around 1km) size. All trajectory points are taken into account for the calculation of the total column footprint maps (top), but only trajectory points between the ground and 500 m agl for the near surface footprint maps (bottom).**

The connection between MO-5 and the other observation sites of the campaign is clearly evidenced by the footprints especially
on 16 March and 1st April (Figure 4C of Rocco et al., 2022). As for PF-1, several trajectories also indicate an origin of the
marine boundary layer. These specific periods will be studied preferentially to follow the lagrangian evolution of the chemical
composition of the air mass.





## 3.2 Dynamical context for a typical cloudy day

An example of the dynamical context of the BIO-MAÏDO measurements is provided on 28 March 2019. Using back-

trajectories and forward-trajectories computed with Meso-CAT, Rocco et al. (2022) and Dominutti et al. (2022a) were able to

show that this situation was typical and highlighted the good connection between the observation sites.





**Figure 4. Meso-NH simulation: Horizontal cross-section at surface for wind direction (color scale) and intensity (vector in m s⁻¹) and liquid water path (mm, in grey) at 02 UTC (a), 08 UTC (b) for March 28. The vertical cross-section shows the liquid water mixing ratio (color scale in g/kg), wind direction and intensity (vector in m s⁻¹), and TKE (green isoline, in m² s⁻²) at 02:00 UTC (c), 08:00 UTC (d) for March 28. (e) MARLEY backscattered signals at DOS-2.**





Figure 4 shows the temporal evolution of the simulated low-level dynamics for the day of 28 March 2019 (500 m domain). The upper figures show the wind direction (color and vector) and intensity (vector size) at the surface at 03 UTC (a) and 08 UTC (b). During this day, the trade wind flow at the surface has a south-east component slightly disturbed by the presence of

cyclone Joaninha located at about 1000 km south-east of Reunion Island. Larger temporal (120 h) and spatial scale (the domain covers 40 to 75 °E) 5 days back-trajectories calculated with CAT and ERA5 ECMWF wind fields show that on 28 March, the air masses arriving to Réunion Island came from the active area of the cyclone whose center is located near 15°S,60°E on 24 March (see supplement material, Figure S1).

Classically, the trade wind flow bypasses the topography of Reunion Island with two zones of wind acceleration off the south-

west and north-east coasts of the island. At the very beginning of the morning (03 UTC), the trade winds return loop is located on the north of the island. This circulation does not penetrate inland (Fig. 4a).

At 08 UTC, the trade winds return loop moved northwest. There is a significant penetration of this surface flow as far as the DOS-2 station. MO-5 is located in a convergence zone between the overflow trade wind flow and this northwesterly counter-flow (Fig 4b). In gray is represented the cloud water content integrated on the vertical. No cloud formation is modeled at 03

UTC apart from a few orographic clouds due to the uplift of the trade winds on the southern flank of the Piton de la Fournaise volcano. Over the BIO-MAÏDO area, the sky is clear (Fig. 4a). At 08 UTC associated to the ascent of the wind flow above the slopes of the Maïdo area, an important formation of clouds is stimulated. All the northwest of the island is concerned by the presence of clouds between 1000 m asl and 2000 m asl (Fig. 4b).

A vertical cross section (Fig. 4c and 4d) has been made in the axis of the red line of Fig. 4a and 4b. This cross section has used

the simulation results from the 100 m horizontal resolution domain at 03 UTC and 08 UTC. At 03 UTC (Fig. 4c), it is noticed that three wind layers have been modeled. Close to the surface, a catabatic wind flow is modeled along the slope. At about 500 m agl, a wind shear is modeled, and the airflow came from the north-west. This last layer is attributable to the return loop of the trade winds. Above, at about 2500 m agl, the wind direction is south-east due to the trade winds overflow above the island. At noon (08 UTC), the anabatic thermal breeze is clearly modeled. This wind regime is added with the trade wind return loop

on a 1 km thick layer (Fig. 4d). This up flow flux reaches 2000 m asl and we find again in the MO-5 area the convergence zone due to the trade winds overflow. As seen before, clouds are simulated over the slopes of the Maïdo area. This presence of clouds concerns almost the entire simulation domain (i.e. between 500 m asl and 2400 m asl). The base of the clouds has reached the surface between 700 m asl and 1900 m asl, therefore over an area covering all the measurement sites except MO-5. Figure 4e shows the time series of the MARLEY backscattered signal at the DOS-2 site for the 28 March until 08:15 UTC

(the system failed afterwards due to a power supply failure). The sky is clear until 06 UTC when the formation of clouds is triggered. The cloud base reaches the surface and the cloud top reaches 2200 m asl. These observations validate the simulated formation of clouds at DOS-2. Due to the dynamical transport above the slopes, these simulation results indicate that the air mass may have been loaded with chemical compounds in the aqueous phase before being evaporated near MO-5.



### 3.3 VOC measurements

During the BIO-MAÏDO campaign three PTR-MS were simultaneously installed for the first time in a tropical area to perform VOC mixing ratio measurements along the slope of the Maïdo road at PF-1, HM-3 and MO-5. In addition, during this campaign, characterization of VOC emissions from tropical and endemic/indigenous and exotic vegetation have been accomplished using solid sorbent cartridges for sampling followed by an analysis with Gas Chromatography – Mass Spectrometer and by Eddy covariance method and PTR-ToF-MS (Time-of-Flight PTR-MS) measurements on HM-3 site.

VOCs measurements were firstly performed during the FARCE campaign in 2015 in Réunion Island (Duflot et al., 2019); concentrations of isoprene were measured at different locations of the island. In this study, the maximal concentrations of isoprene were measured in the 100 to 200 pptv range in the tropical forest site (Bélouve forest). From 2018 to 2021, a PTR-MS was installed at MO-5 for continuous VOCs measurements as part of the OCTAVE project. These measures have been used in the Verreyken et al., (2020) that addressed the impact of long-range biomass burning at remote sites at MO-5 and an

overview of the 2-year campaign is presented in Verreyken et al., (2021). In this last study, high levels of BVOC have been observed and isoprene concentrations reached up to 500 pptv. In addition, during the OCTAVE project, a second PTR-MS was positioned in the Bélouve forest (20.9° S, 55.3° E, 7 m asl.) and at Le Port (21.06° S, 55.5° E, 1498 m asl) sites for 10 days each in April-May 2018. In a study dedicated to the analysis of formaldehyde sources and origin at MO-5 using these additional measurements, Rocco et al. (2020) found that most of formaldehyde is formed from biogenic secondary compounds

(oxidation of biogenic VOCs with 37% in average).

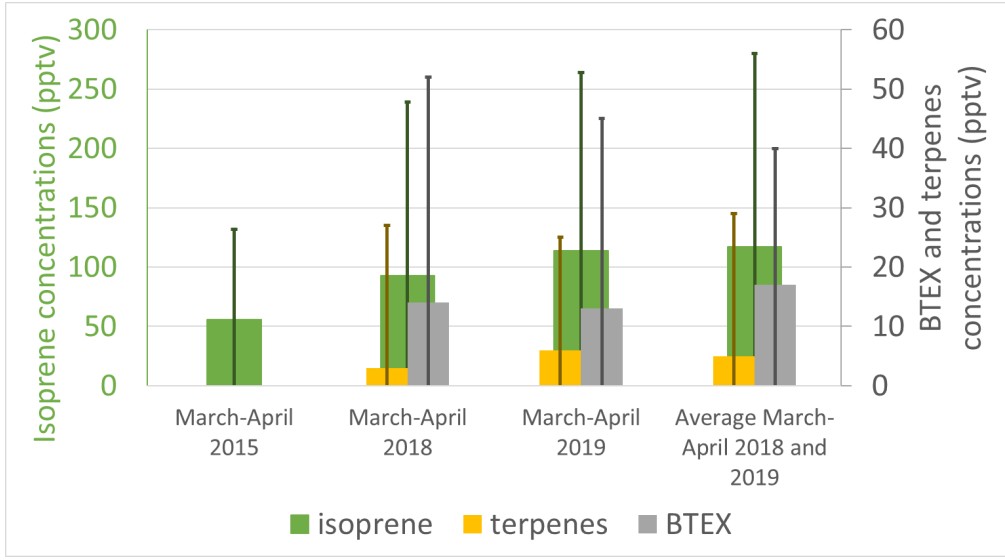

**Figure 5: Isoprene, terpenes and BTEX mixing ratio (pptv) for the FARCE (2015, Duflot et al., 2019), OCTAVE-1 (2018, Rocco et al., 2020), BIO-MAÏDO (2019, Rocco et al., 2022) and OCTAVE-2 (March-April 2018-2019, Amelynck et al., 2021) at the Maïdo Observatory.**

Figure 5 compares averaged mixing ratios of isoprene, terpenes and BTEX (sum of benzene, toluene, ethylbenzene and xylenes) at MO-5 for four datasets corresponding to FARCE, OCTAVE-1 (first – 2018 datasets) BIO-MAÏDO, and OCTAVE-





2 (2018 and 2019 datasets) campaigns. Dominant mixing ratios are observed for isoprene all over the different campaigns with averaged concentrations of 95 ± 133 pptv. Low mixing ratio of terpenes and BTEX are measured at MO-5 showing low influence of anthropogenic emissions and low emissions of monoterpenes by local vegetation. Level of isoprene are higher for

BIO-MAÏDO and the second set of OCTAVE than for FARCE and the first set of OCTAVE. For FARCE campaign, terpenes and BTEX are not available. Terpenes are the highest for BIO-MAÏDO and BTEX are the smallest but with comparable levels with other datasets.

The coupling of VOCs chemistry and dynamics measured during BIO-MAÏDO campaign was investigated to better understand the role of dynamics in the distribution of VOCs (Rocco et al., 2022). This new and first approach combined cover land

footprint and backward trajectories. It provided information on the nature of the ground-surface influencing the air masses during the few days and hours before the air mass arrives at the sampling sites (PF-1 and MO-5). The variability of VOC concentrations along the slope was also analyzed. The origin of air masses greatly varied among days. They showed differences in forward and backward trajectories coming from PF-1 to MO-5 with air masses more or less advected from the down-slope areas. Many days were marked by a frequent oceanic air mass origin (up to 50%) with high concentrations of methanol and

acetone. Ratio of isoprene oxidation products to isoprene concentration have been calculated. Calculated ratios were in average 0.44 ± 0.42 at MO-5 and 1.11 ± 1.59 in PF-1. A lower ratio at MO-5 indicates recent emissions of isoprene, and therefore a major contribution from the local vegetation, which has not yet had time to oxidize to secondary compounds.

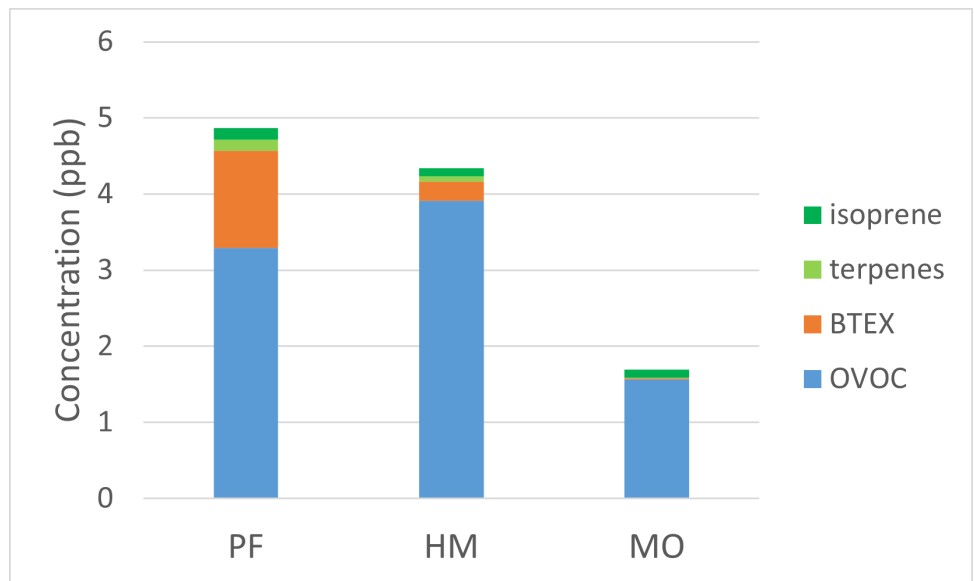

**Figure 6. Average concentration of isoprene, terpenes, OVOC and BTEX concentration in ppb at Petite France (PF-1), Hotel du**
**Maïdo (HM-3) and at the Maïdo observatory (MO-5).**

Figure 6 shows the average composition of isoprene, terpenes, OVOC (sum of methanol, acetaldehyde, acetone and methyl ethyl ketone/MEK) and BTEX mixing ratios at PF-1, HM-3 and MO-5 averaged during the whole BIO-MAÏDO campaign. A spatial gradient in total VOC mixing ratios is observed from PF-1 to MO-5 with a decrease by a factor of 2.5 between PF-1



and MO-5. OVOCs are the major contributors to total VOC burden (> 50% in vol); while the contribution is getting lower for primary biogenic and anthropogenic VOC from PF-1 to MO-5 where OVOC is dominant. This gradient depends on the distance to the main primary sources (i.e. vegetation and traffic), air mass history during its transport (cloud presence, surface characteristics) and air mass aging.

During BIO-MAÏDO campaign, mixing ratios of isoprene measured at the 3 sites are between 100 and 600 pptv. Average isoprene mixing ratios for the period was $0.16 \pm 0.12$ ppbv, $0.11 \pm 0.10$ ppbv and $0.12 \pm 0.15$ ppbv at PF-1, HM-3 and MO-5 respectively. Terpene's mixing ratios are 20 times higher in PF-1 than MO-5, reaching a value of 140 pptv. As PF-1 is a mixed rural and urban site, the sources of terpenes are more abundant in this site than in MO-5. At HM-3, averaged concentrations of $\alpha$-pinene, $\beta$-pinene and limonene were respectively $0.06 \pm 0.04$ ppbv, $0.01 \pm 0.01$ ppbv and $0.09 \pm 0.05$ ppbv. As marker of anthropogenic emissions, BTEX are also present in higher level at PF-1 ($1.27 \pm 0.67$ ppbv) than at MO-5 and HM-3. Dilution and oxidation processes explain the decreasing levels due to the increased distance to the source. Other hypotheses must be considered. Despite BTEX are poorly soluble in water, BTEX was detected in every cloud water samples with a mean concentration of 4.2 nM (Dominutti et al., 2022) showing that clouds act as a sink for aromatic compounds. Another potential sink is the deposition of BTEX on the leaf cuticle through gaseous deposition (Molina et al, 2021). Finally, Rocco et al., (2022), showed that PF-1 and MO-5 was not dynamically connected every day during the campaign. Concerning the OVOC, the species in higher mixing ratios for the three sites is methanol with an average mixing ratio of $2.16 \pm 0.89$ ppbv, $2.79 \pm 1.10$ ppbv and $0.82 \pm 0.35$ ppbv at PF-1, HM-3 and MO-5 respectively. As this compound is primarily emitted from terrestrial plant during the growth and the decay stages (Bates et al., 2021 and references therein), this can explain the highest mixing ratio observed at HM-3.

## 3.4 Aerosol measurements

An online ToF-ACSM was used to determine the chemical composition of non-refractory- $PM_1$ (NR-$PM_1$) aerosol at MO-5, providing mass concentrations for organic, sulphate, nitrate, ammonium and chloride species. At MO-5 this instrument operated continuously from March 13th to April 2nd and showed an average NR-$PM_1$ mass concentration of $4.6 \pm 6.2$ µg m$^{-3}$ with a strong diurnal variability. Daily mass concentrations reaching up to 25 µg m$^{-3}$ were observed at the start and the end of the field campaign, while nighttime concentrations, when the site was most likely in the free troposphere, were close to the limit of detection of the instrument. These measurements were coherent with aerosol number concentrations measured by DMPS that showed similar diurnal profiles with typical signatures of new particle formation on a daily basis (Rose et al., 2019).

The NR-$PM_1$ were dominated by $SO_4^{2-}$ (57.3%), followed by organics (23.3%), $NH_4^+$ (14.2%), and $NO_3^-$ (2.2%) (Dominutti et al., 2022b). The high concentration of sulfate containing particles and the low concentration of $NH_4^+$, show that forms of sulfate, other than ammonium sulphate (($NH_4)_2SO_4$), were sampled, likely acidic aerosol such as $NH_4HSO_4$ or eventually in the form of organosulphates (Brito et al., 2018).



The contribution of different organic species to the total organic mass concentration was determined using positive mass factorization (PMF), with the source finder (SoFi) tool (Canonaco et al., 2013). Three factors were resolved using PMF analysis on the entire organic matrix from m/z 1 to 150; a more oxidized organic aerosol (MOOA) (75%), a primary organic aerosol (POA) (18.5%), and an isoprene derived organic aerosol (IEPOX-OA) (11%).

During the second part of the field campaign, air masses were exposed to aqueous phase processing (as determined using the results of Meso-NH model). Using this information, aerosol chemical composition and physical properties were compared under both clear and cloudy conditions. A clear shift in the aerosol size distribution was observed (an increase by 15% of Aitken and accumulation mode aerosols under cloudy conditions), as well as a shift in the organic aerosol chemistry with increases in MOOA, in oxalic acid concentrations and in sulfate aerosols in the $PM_{10}$ offline filters. These observations together

with model estimates of in-cloud processing of aerosols suggest that oxidation of gaseous precursors, and primary organic aerosol species and other aqueous phase processing have a significant impact on the sources of organic aerosol (notably oxalic acid), and on aerosol physical properties (Dominutti et al., 2022b).

Additionally, $PM_{10}$ aerosols were simultaneously sampled by offline filters at MO-5 and PF-1 during the whole field campaign. Figure 7 present the average concentrations of different $PM_{10}$ components observed at MO-5 and PF-1 during the BIO-MAÏDO

field campaign, split between daytime and nighttime.

The average $PM_{10}$ concentrations showed significant differences between the MO-5 and PF-1 sites. The main discrepancies were observed for total organic matter (OM) concentrations, which were higher at PF-1 (3480 ng $m^{-3}$) than at MO-5 (1506 ng $m^{-3}$) by a factor of 2.3 (Fig. 7a and 7b). In addition, higher concentrations at PF-1 were also observed for $Na^+$ and $NO_3^-$ (by a factor of 2) and $Cl^-$ and EC (by a factor of 3.6 and 4.5, respectively) (Fig. 7a and 7b). These differences could be related to the

distance of the sites from emission sources, as is the case of marine origin ions $Cl^-$ and $Na^+$, and the traffic-related ones, EC and $NO_3^-$. On the other hand, sulfate and ammonium being associated with long range transport, do not differ significantly in the average concentrations of the sites. A second large difference comes from the fact that MO-5 is in free tropospheric conditions during the night, leading to much larger differences in the day/night ratios at MO-5 than at PF-1. Notably, OM at PF-1 does not show a substantial difference between daytime and night-time (2.01 and 1.81 μg $m^{-3}$, respectively), however, its

concentrations were dissimilar at MO-5 (0.90 and 0.35 μg $m^{-3}$, respectively).

The OM composition was also variable between the sites and also on a day/night basis (Fig. 7c). As expected, sugars alcohols and monosaccharides had higher concentrations at PF-1 than MO-5, by a factor of 3 to 10. Mean concentrations at PF-1 were determined by arabitol (49 ng $m^{-3}$), levoglucosan (38.6 ng $m^{-3}$), mannitol (35.5 ng $m^{-3}$), and glycerol (19.3 ng $m^{-3}$). At MO-5, a similar profile is observed but differently controlled by levoglucosan (8.8 ng $m^{-3}$), mannitol (6.5 ng $m^{-3}$), trehalose (4.9 ng

$m^{-3}$), and arabitol (4.4 ng $m^{-3}$). Interestingly, higher concentrations of sugars were observed at night at PF-1 (Fig. 7c), suggesting that environmental conditions (such as temperature and humidity) can have a role in the emission processes of these compounds by natural sources (e.g., soils, bioaerosols, plants and fungal spores). Zhang et al., (2010) found that arabitol and mannitol in $PM_{10}$ showed significant correlations with relative humidity and air temperature, suggesting a wet emission mechanism of biogenic aerosol in the form of fungal spores in a tropical rainforest. The sugar alcohols, mannitol and arabitol,




are common energy reserves in fungi and are produced in large amounts by many fungi (Golly et al., 2019, Zhang et al 2010, Bauer et al 2008). In contrast, levoglucosan, a degradation product from biopolymers, is known as a good molecular tracer of biomass burning in the literature (Simoneit et al., 2002). However, levoglucosan concentrations observed in our study are more likely to be due to domestic biomass burning (e.g. cooking emissions) rather than forest fires (not reported in the area during the campaign). Thus, our results show a strong contribution of biogenic sources on $PM_{10}$ samples such as fungi spores, soils, and microorganisms and to a lesser extent the contribution from biomass burning aerosols.

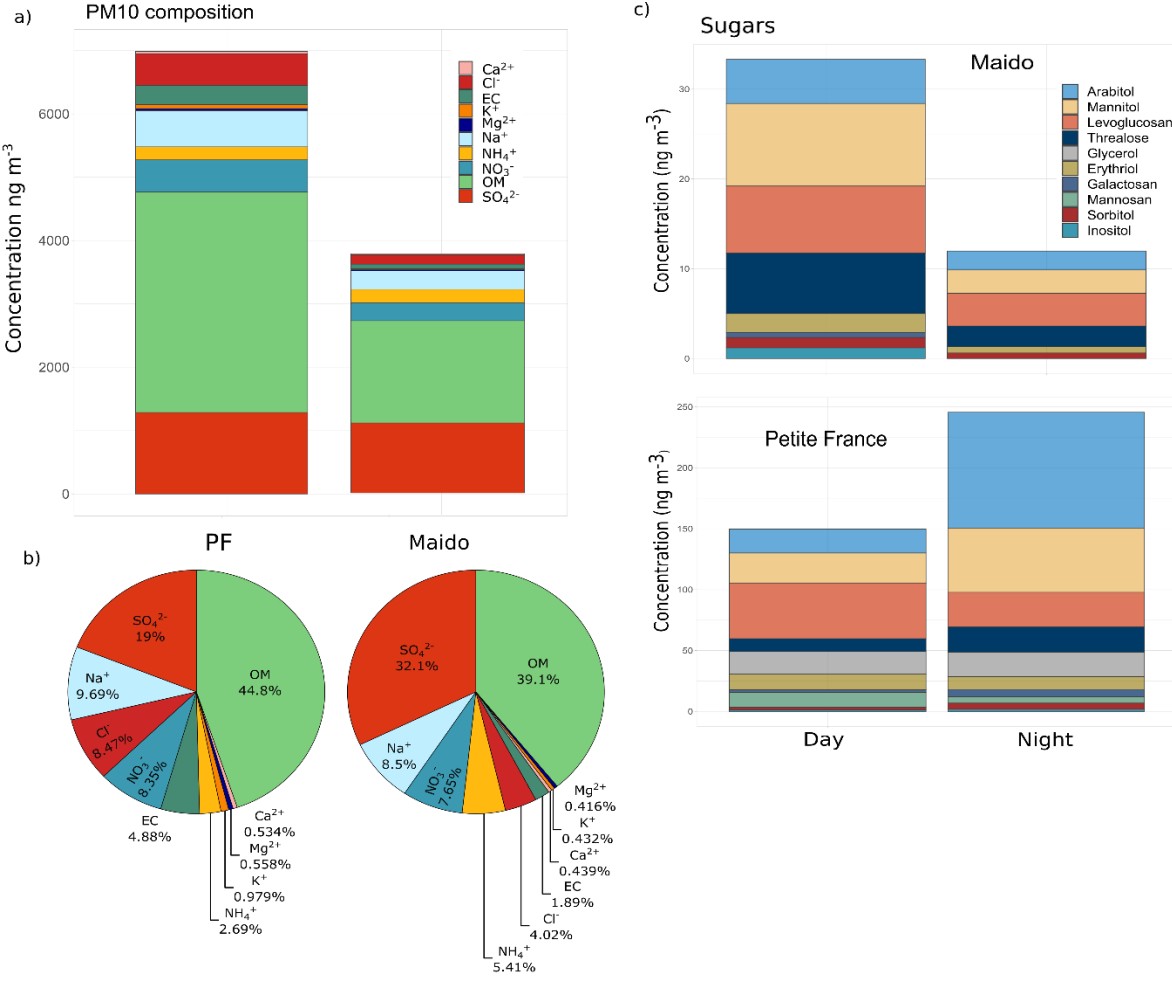

**Figure 7. (a) Average PM10 composition observed at Petite France (PF-1) and Maido Observatory (MO-5), (b) Relative contribution of PM10 components during daytime observations and c) average sugars concentrations at PF-1 and MO-5 sites during day and nighttime.**





**Figure 8. (a)** Time series evolution of organic acid concentrations observed at MO-5 and PF-1 and **(b)** average concentrations observed at each site during day and night observations.

Seventeen organic acids concentration were measured at both sites (Fig. 8). Contrarily to what was observed for ions and

sugars, higher mean concentrations of total organic acids were obtained at MO-5 (95.3 ng m$^{-3}$) than PF-1 (56.5 ng m$^{-3}$). The largest contribution at both sites was from oxalate, which presented average daytime concentrations of 63 ng m$^{-3}$ at MO-5 and 15 ng m$^{-3}$ at PF-1. Oxalic acid is the most abundant and ubiquitous dicarboxylic acid (Kawamura and Bikina, 2016) and is commonly associated as secondary organic tracer formed from the photochemical/aqueous oxidation of many organic precursors (Ervens et al., 2011). As discussed by Dominutti et al., (2022b), the large contribution of this acid at MO-5 suggests

the impact of more oxidized aerosols transported over long distances. Organic acids also include the presence of other dicarboxylic acids, such as malonic, malic and succinic, species typically observed in other and similar environments (Kawamura and Bikina, 2016, Golly et al., 2019, Cheng et al., 2013, Wang et al 2006). The average daytime concentrations of malonic were similar at both sites (15.6 and 12.1 ng m$^{-3}$ at PF-1 and MO-5, respectively), however, succinic (5.9 and 9.6





ng m$^{-3}$ at PF-1 and MO-5, respectively) and malic (10.4 and 4.2 ng m$^{-3}$ at PF-1 and MO-5, respectively) exhibited some small differences between sites. These acids can be emitted into the atmosphere from various anthropogenic (e.g. vehicular, biomass burning) and natural (marine aerosols) sources, but there are mainly produced in the atmosphere by several photochemical reactions of their organic precursors. The involvement of the photochemical process in the production of those acids can be evaluated by the mass ratio of malonic to succinic acid (<1 for photochemically aged aerosols, Kawamura and Sakaguchi, 1999). Our observations show average malonic/succinic ratios of 2.62 and 1.51 at PF-1 and MO-5, respectively, confirming the presence of photochemical aged aerosols in Réunion Island.

Overall, the chemical profiles and PM$_{10}$ concentrations show that the MO-5 and PF-1 sites are rather disconnected during most of the field campaign, especially at night. The results reveal that different environmental conditions and atmospheric dynamics have an impact on the spatial distribution and composition of aerosols in Reunion Island.

### 3.5 Cloud chemistry analysis

During the BIO-MAÏDO campaign, 14 cloud samples were collected at PO-4 and characterized by physico-chemical and microbiological analysis. This section is devoted to summarize the main results described in detail in Dominutti et al. (2022a) and to present the ongoing works.

Data obtained with the cloud droplet probe (CDP) reveal that clouds collected on the slope of the mountainous island present low water content with LWC values of $0.07 \pm 0.04$ g m$^{-3}$ on average. Those values are more representative of LWC reported for fogs than the higher ones reported for marine clouds (values closer to 0.2 to 0.4 g m$^{-3}$), such as those sampled at Puerto Rico (Gioda et al., 2013) or Cape Verde (Triesch et al., 2021). This is to the atmospheric dynamical circulation that leads to the formation of clouds with low LWC over this part of the island (see section 3.2 and Duflot et al., 2019). Concerning the size of the droplets, the D$_{eff}$ for the 14 cloud events is rather small, with an average value of $13.7 \pm 1.51$ µm (Figure 9a).

The main inorganic ions have been quantified in priority since they are used as tracers of various natural and anthropogenic sources (Deguillaume et al., 2014). In line with the low LWC, the total concentrations of these ions are little diluted and therefore high, with concentrations ranging from 600 to 4370 µmol L$^{-1}$. As expected, for all the cloud events, a major influence of ions from marine origin is found, confirming the contribution of sea salt to the cloud formation (Na$^+$: $490 \pm 399$ µmol L$^{-1}$; Cl$^-$: $434 \pm 370$ µmol L$^{-1}$) (Figure 9b). Those amounts are similar to those observed for other marine sites. Nitrates are the third ions in relative contribution, with a concentration of $239 \pm 168$ µmol L$^{-1}$. These elevated concentrations may be linked to local anthropogenic sources (uptake of NO$_x$/nitric acid from the gas phase into the droplets or dissolution of nitrate from aerosols). This additional anthropogenic fraction is confirmed by the presence of sulfate in a significant quantity ($118 \pm 44$ µmol L$^{-1}$). The measured SO$_4^{2-}$/Na$^+$ ratio is much higher than the standard sea-salt molar ratio by a factor of 2.8 on average, confirming anthropogenic contribution to the sulfate amount. The contribution to the sulfate concentration of volcanic emissions through the dissolution of SO$_2$ in cloud droplets and oxidation to form sulfates cannot be ignored, even if no eruption was reported during the sampling period. Ammonium levels ($123 \pm 43$ µmol L$^{-1}$) are comparable to the observations conducted for remote continental sites, indicating plausible terrestrial/agricultural inputs. The concentration of these compounds determines cloud





water acidity, leading to acidification (i.e., $SO_4^{2-}$, $NO_3^-$) and/or basification (i.e., $NH_4^+$, $Mg^{2+}$, $K^+$, $Ca^{2+}$) together with the $CO_2$ dissolution from the gas phase. The pH of the cloud samples does not vary a lot, with values ranging from 4.7 to 5.5. Finally, trace metals have been quantified and present very low concentrations. Mg and Zn, which have natural origin such as sea salt,

present the most important concentrations, followed by Cu, Fe, Mn, Ni, Sr, and V. These amounts are in the same range as previous studies performed in marine environments (Fomba et al., 2013, 2020) or influenced by marine emissions (Bianco et al., 2017). Iron speciation has been evaluated with Fe(II)/Fe(II)+Fe(III) ratio equal to 52 ± 22 %. This suggests an efficient conversion of Fe(III) to Fe(II) and possible complexation of Fe(III) with organics, leading to its stabilization under this redox form. Nonetheless, the effect of iron on the oxidative budget is expected to be low due to low Fe concentrations (0.88 ± 0.19

µmol $L^{-1}$ on average).

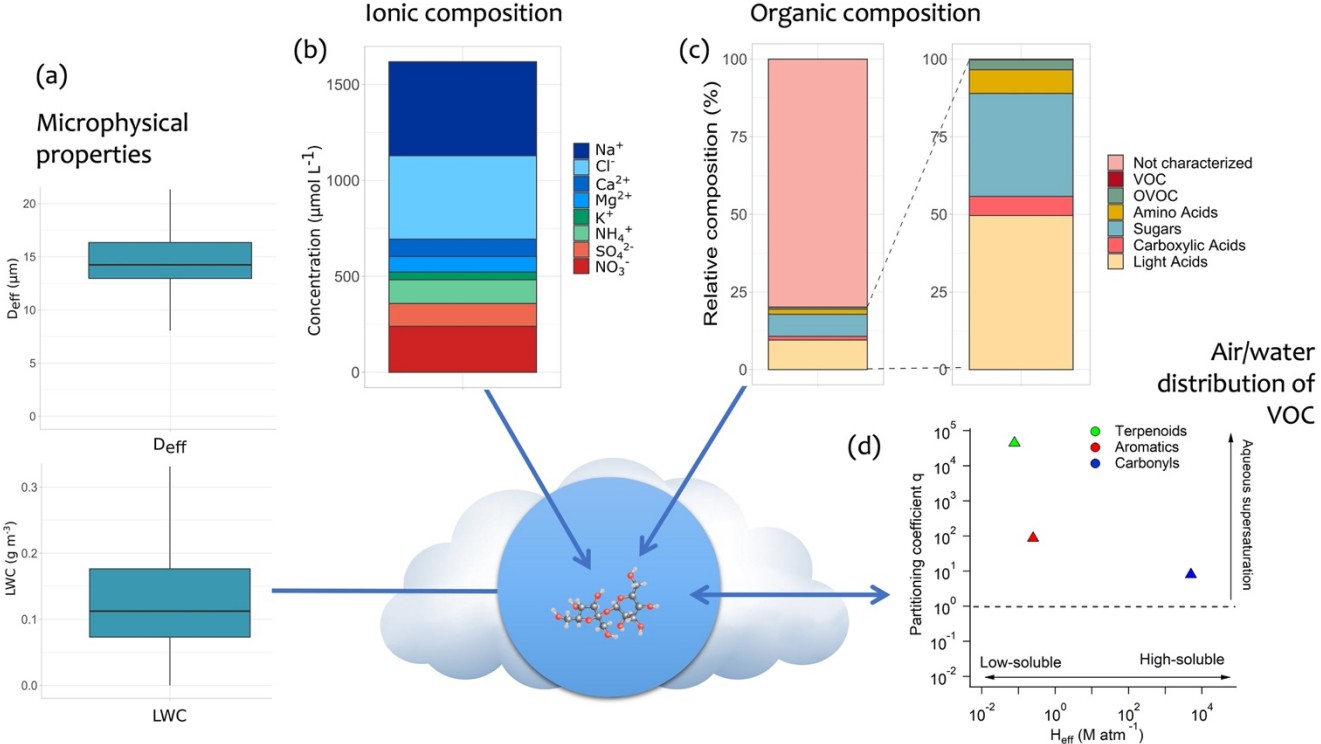

**Figure 9. Summary of the main results of the cloud water chemical characterization: (a) Microphysical properties (LWC and Deff), (b) concentration of the main inorganic ions, (c) relative organic composition, (d) partitioning coefficient q of VOC between the gaseous and aqueous phase as a function of effective Henry's law constant (Heff).**

Dissolved organic compounds have also been intensively investigated during the campaign. The average concentration of dissolved organic carbon (DOC) is equal to 25.5 ± 19.2 mg C $L^{-1}$ that is much higher than values reported for cloud waters sampled in marine environments such as at the southern Pacific Ocean (Benedict et al., 2012), Puerto Rico (Reyes-Rodriguez et al., 2009), southern Asia (Stahl et al., 2021) or the puy de Dôme station for clouds under the marine influence (Deguillaume et al., 2014). Like for inorganics, this indicates additional inputs of DOC other than sea-related ones. Among the quantified



compounds, organic acids and sugars contribute on average to a significant fraction of the DOC (around 18%) (Figure 9c). Light carboxylic acids, such as formic and acetic acids, are dominant compounds and present concentrations higher than those observed in marine environments. Results indicate that their concentration in the aqueous phase is mainly due to mass transfer from the gas phase, in which they are emitted by anthropogenic and biogenic primary sources. The most concentrated dicarboxylic acids are lactic and oxalic acids, resulting from the reactivity in the aqueous phase or from the dissolution of the

CCN. Reported concentrations of dicarboxylic acids are rather low compared to other sites, probably because clouds freshly form over the mountain slope, which does not allow their efficient production by aqueous reactivity. Sugars are also ubiquitous in all our samples and derive mainly from the water-soluble fraction of the CCN. The aqueous concentration of sugars is equal on average to $22.2 \pm 15.4$ µmol $L^{-1}$, and the calculated atmospheric concentration ($121.3 \pm 69.6$ ng $m^{-3}$) is important compared to previous aerosol studies (Verma et al., 2018; Zhu et al., 2015). This can be explained by the importance of biogenic

emissions at Réunion Island but potentially also by the production by microorganisms in cloud water. Identically to sugars, amino acids are issued from biogenic production. They have been measured with total concentrations of free amino acids (TCAA) varying from 0.8 to 21.1 µmol $L^{-1}$ (average: $4.6 \pm 5.5$ µmol $L^{-1}$), representing 1.6% of the DOC in average. These values are higher than those reported at a marine site in Cape Verde by Triesch et al. (2021) and much higher than those measured at the puy de Dôme station (Renard et al., 2022). This fact can be explained by the surrounding potentially important

sources (sea surface but also vegetation). Among the 15 amino acids detected, Serine, Alanine, and Glycine are dominant; a plausible explanation is related to their high atmospheric lifetimes due to low reactivity with hydroxyl radicals and their preponderance in biological matrices. Finally, carbonyls (OVOC) and low solubility VOC have been investigated. The average concentration of carbonyls is $3.5 \pm 1.7$ µmol $L^{-1}$ (on average: 42.0% of formaldehyde, 14.2% of hydroxyacetaldehyde, 11.3% of acetaldehyde, 10.4% of acetone, 9.8% of glyoxal, 6.9% of hydroxyacetone and 5.3% of methyl glyoxal), representing 1%

of the DOC on average. This global amount is rather low compared to other studies, and the formaldehyde to acetaldehyde ratio suggests a contribution of vegetation emissions. Terpenoids (α-pinene, β- pinene, and limonene) and isoprene have been detected together with primary aromatics (benzene, toluene, ethylbenzene, xylenes) in nmol $L^{-1}$ range, contributing to 0.35% of the DOC. Despite their low concentrations, they are of interest since they are tracers of emissions and allow for evaluation air/droplet partitioning (see below).

The contribution of the organic compounds targeted in this project represents around 20% on average of the DOC concentration, reaching up to 35% for one specific cloud. This reveals the complexity of the cloud organic matter that has been recently highlighted by high-resolution mass spectrometry (Cook et al., 2017; Bianco et al., 2018; Sun et al., 2021). Thousands of molecular formulae have been detected in cloud water, and their presence and concentration are related to the influence of primary sources and atmospheric processing. The analysis by FT-ICR MS requires at least 50 mL of cloud water, which is

thereafter pre-concentrated and desalted by solid-phase extraction. Three samples (R8, R9, and R10B) presented enough volume to be analyzed with this technique. Two complementary ionization sources focused the attention on different organic compounds: electrospray ionization in positive and negative polarities enabled the identification of polar and semipolar compounds, while atmospheric pressure photoionization (APPI) revealed the presence of less polar compounds, probably





related to lipids and terpenes. The assigned molecular formula retrieved by high-resolution mass spectra does not enable the
attribution of a structural formula. Nevertheless, based on the elemental composition, considering the number of carbon,
oxygen, nitrogen, hydrogen, and sulfur atoms, the molecular formula can be classified into compounds of biogenic and
anthropogenic origin, such as lipids, carbohydrates, proteins or unsaturated hydrocarbons and condensed aromatics,
respectively. This analysis offers the possibility of getting information on the DOC fraction not characterized by previously
presented targeted analysis. The detailed presentation of FT-ICR MS results will be reported (work in progress). In addition
to the chemical characterization, specific attention has been paid to the distribution of VOC between the gas and the aqueous
phase in the cloudy atmosphere and to the environmental variability of the chemical composition. For OVOC, a small deviation
from Henry's law equilibrium has been observed. However, high supersaturation in the aqueous phase is measured for low
soluble biogenic and anthropogenic compounds as previously described in van Pinxteren et al. (2005) and Wang et al. (2020)
(Figure 9d). Possible explanations are interactions of these compounds with dissolved or colloidal matter or adsorption at the
air-water interface.

A statistical analysis has been performed combining cloud chemical data with back-trajectory calculations derived from Meso-
CAT associated with the Corine 2018 land-cover database. This work shows that air mass origin and microphysical variables
cannot explain the evolutions observed in cloud chemical composition. This reveals the complexity of interconnected processes
occurring on the mountain slopes (i.e, emission sources, multiphasic transfer, and chemical processing in clouds).

**3.6 Biological measurements: cloud water and aerosols**

The diversity of culturable bacteria from clouds collected at PO-4 during this campaign are reported, along with others, in
Charpentier et al. (submitted, 2022). These included 105 distinct strains, most of which (58%) were affiliated with
Proteobacteria (39% Gammaproteobacteria, 12% Alphaproteobacteria and 7% Betaproteobacteria), followed by
Actinobacteria (34%) and Firmicutes (7%). The most represented species comprised *Stenotrophomonas*, *Pseudomonas* and
*Acinetobacter* in Gammaproteobacteria, *Microbacterium* and *Curtobacterium* in Actinobacteria, *Bosea* and *Sphingomonas* in
Alphaproteobacteria, and *Bacillus* in Firmicutes. This is a common pattern for viable bacteria in atmospheric samples (e.g.,
Vaïtilingom et al., 2012).

Bacteria diversity profiling by high-throughput sequencing from 74 aerosol samples collected at PF-1 and Maïdo indicated the
presence of 8,437 $OTU_{0.03}$ (operational taxonomy units clustered at 97% sequence similarity, i.e. ~ prokaryotic species level
see Amato et al. (2017) for details) in total, of which 6,935 were affiliated at >95% identity to know sequences in databases.
The vast majority of these (99.6%) were bacteria, the remaining being attributed to Archaea. As for culturable bacteria from
clouds, the phylum Proteobacteria dominated (29.3% of the sequences), before Firmicutes, Actinobacteria and Bacteroidetes
represented each by ~16% of the sequences. Among others, Cyanobacteria, Planctomycetes and Acidobacteria notably
contributed each 3% to 4% of the sequences. This composition is consistent with airborne bacteria at other places of the planet
using similar methods (e.g. Amato et al., 2017; Péguilhan et al., 2021).



## 4. Discussion

Online measurements of VOC during the campaign show, as expected, the presence of BVOC dominated by isoprene along the slope of the Maïdo and at the observatory during the daytime. The effect of cloud on mixing ratios of isoprene and its oxidation products is clear looking at their diurnal variation for cloudy days (See Fig. 4 in Rocco et al., 2022). It is well-known that clouds induce less efficient emission of isoprene due to the decrease of solar radiation and temperature (e.g. Guenther et al., 1993), but the scavenging of isoprene and its oxidation products in cloud droplets could also contribute to their decrease. Dominutti et al. (2022a) have shown that isoprene has been detected in almost all the samplings of cloud water during the campaign (at concentration of about a few dozen nmol $L^{-1}$). Furthermore, despite BTEX are poorly soluble in water, BTEX were detected in every cloud water samples with a mean concentration of 4.2 nmol $L^{-1}$ showing that clouds act as a sink for aromatics compounds. However, even if the level of oxidation products from isoprene decreases from PF-1 to MO-5 for days where both sites are dynamically connected, VOC measurements do not allow determining the importance of the photochemistry versus the dilution and the deposition on the air mass composition sampled at MO-5 (Rocco et al., 2022).

The analysis of PM$_{10}$ filters shows that, even if the concentration of organic matter (OM) is lower at MO-5 than at PF-1, the concentration of dicarboxylic acids is the highest at MO-5 especially in the second part of the campaign where oxalate is detected only at MO-5 (Fig. 8a), and which was cloudier. The PMF analysis of the submicron aerosols at MO-5 show that the more oxidized organic fraction of aerosols (MOOAs) is the dominant part of the organic aerosol, and its contribution increases during the second part of the campaign (Dominutti et al., 2022b). These results seem to underline the possible effect of cloud processing on the organic composition of particles sampled at MO-5. An analysis combining observations of PM$_{10}$ composition, aerosol size distribution, contribution of PMF factors and results of simulated clouds along the slope from Meso-NH model highlights the role of cloud processing on aerosols sampled at MO-5 (Dominutti et al., 2022b). Observations showed a shift to larger diameter of the aerosol size distribution (15% for Aitken and accumulation modes) and an increase of 10% of the contribution of sulfate and MOOAs in the chemical composition of submicron particles when cloud processing occurs during the daytime (Dominutti et al., 2022b).

The database from the BIO-MAÏDO campaign is a unique opportunity to better understand the contribution of multiphase atmospheric chemistry on SOA formation. However, deep analysis of the database does not allow to quantify this contribution even if it shows clear evidence of cloud processing on the OM composition of particles sampled at MO-5. Simulations with the explicit cloud chemistry model CLEPS is ongoing to investigate organic matter processing by cloud. A case study has been selected for this simulation, the 28th of March 2019 was chosen for several reasons: the amount of cloud water sampling was sufficient for allowing deep chemical composition analysis (Dominutti et al., 2022a), the observed cloud that day is typical of a cloud forming on the slope at the end of the morning and evaporating before arriving at the MO-5 (see section 3.2), the operation of instruments was almost complete and there is evidence for sampling of air mass advected along the slope at MO-5 (Rocco et al., 2022). The CLEPS simulation is driven by meteorological, and cloud microphysical parameters extracted from a trajectory coming from the Meso-NH smallest domain (100m horizontal resolution) for this specific cloud event. The





contribution of biodegradation on the cloud processing will be assessed thanks to a recent development in the CLEPS model
(Pailler et al., 2023). Biodegradation rates considered in the model have been determined using microbial strains that have
been isolated from the puy de Dôme station. But this is not problematic since the comparison of the profiles of the phylum
level distributions of bacteria isolates from clouds at Réunion Island and at puy de Dôme shows close similarities (Charpentier
et al., submitted, 2022). Microorganisms' metabolic activities could be even more efficient at Réunion Island due to more
elevated temperature, and this can be parametrized in the model.

In parallel with this 0d modeling, based on the 3D Meso-NH simulation made for the entire campaign, other Meso-NH
simulations were made for the specific period from March 28 to 30. El Gdachi et al. (in prep), combined the size and
composition of the observed aerosols to couple them to a two-moment microphysical scheme. This new detailed simulation
also uses a very high vertical resolution (1m near the surface) to accurately represent anabatic and katabatic thermal circulations
and the formation of clouds on the topography. Mouchel-Vallon et al. (in prep) carried out 3D studies combining gas-phase
chemistry and a detailed inventory of BVOC sources (100m) to study the modes of transport and oxidation of organic
secondary aerosol precursors. Estimations of emissions of BVOC at HM-3 (see section 2.4) are used to calibrate and validate
simulated emission fluxes. These both high-resolution 3D simulations (100m horizontal resolution and 1m vertical resolution
near the surface) are able to correctly represent the life cycle of clouds and the main thermal circulations on slopes (anabatic
and katabatic) as well as the source regions and isoprene oxidation mechanisms (El Gdachi et al, in prep; Mouchel-Vallon et
al., in prep). Finally, a specific simulation including gas-phase, aerosols and cloud chemistry with Meso-NH will be performed
for the case studied with CLEPS. CLEPS simulation will help to select the dominant chemical pathways in aqueous phase to
be considered and/or added in the Meso-NH chemical mechanism. 3D modelling allows considering complex effects of
dynamics, deposition, emissions, and photochemistry on the air mass arriving at MO-5.

## 5. Conclusion

The BIO-MÄIDO campaign was dedicated to the observation of the effect of cloud on chemical composition of aerosols and
especially the organic part in an environment dominated by natural and biogenic emissions. The Maïdo observatory, which
was inaugurated at the end of 2012, was a unique opportunity to fulfil this objective because of its set of instrumentation
(https://www.osureunion.fr/les-stations-dobservation/opar/parametres-mesures/, last access: 20 June 2023) and because of its
geographical situation: tropical environment, quasi-daily cloud formation on the slope down the observatory for a specific part
of the year, isolated island with only reduced local anthropogenic influence, and endemic forest on the slope down the
observatory. The strategy of the campaign and the choice of the deployed instrumentation was worked out to get the needed
parameters to understand the chemical composition of aerosols sampled at the observatory and the signature of the cloud
influence on it. The database from the campaign is original, combining dynamical, microphysical, chemical, biological, and
particles' size parameters (droplets and aerosols).



The study of the mixing boundary layer air masses advected by thermal breezes at MO-5 during the daytime shows two preferred trajectories routes both corresponding to the return branches of the trade winds associated with the up-slopes thermal breezes. The first preferred route for airmass trajectories passed through the south allowing air masses to pass over the forests located between 1000 m and 1500 m asl in the south-west of the island. The route of the second set of trajectories is going up the western flank of the island, also indicating a passage in the marine boundary layer. A detailed analysis based on a high-

resolution Meso-NH simulation is made for the 28[th] of March. This analysis shows an important formation of clouds on the slope of the Maïdo at the end of the morning associated to the anabatic thermal breeze added with the trade winds return loop. These clouds evaporate before arriving at MO-5 indicating that aerosol particles measured at MO-5 can be thought to have undergone cloud processing during its transport on the slope.

The analysis of VOC measurements shows a highest mixing ratio of BTEX, isoprene and terpenes at PF-1. However, OVOC

is highest at HM-3 and the highest contributor of VOC mixing ratio at MO-5. The diminution of BTEX from PF-1 to HM-3 to MO-5 is a signature of the decrease of the influence of anthropogenic emissions along the slope to the Maïdo, as expected. The study of the chemical composition of particles at PF-1 and MO-5 during the daytime shows the presence of more oxidized organic aerosol at MO-5 and a higher concentration of oxalic acids at MO-5 than at PF-1. Both results indicate the presence of photochemical aged aerosols at MO-5, potentially impacted by cloud processing depending on the day and the trajectories

of air masses arriving at MO-5. The analysis of cloud chemical composition allows a thorough identification of organic compounds in cloud water. Despite this, around 80% in average of dissolved organic compounds in cloud water are undefined highlighting the complexity of the cloud organic matter.

These results, obtained from analysis of the BIO-MAÏDO campaign database, must be completed by numerical modeling to answer the three main objectives of the BIO-MAÏDO project i.e. understand which are the main formation pathways of SOA

in humid tropical atmosphere (gaseous phase versus aqueous phase); improve multiphase processes leading to SOA formation in a 3D model; examine whether the presence of bacteria in aqueous phase could contribute to SOA formation. To assess those objectives, simulations with the explicit cloud chemistry CLEPS model and the 3D Meso-NH are underway. Results from CLEPS will help to develop more complete chemical mechanisms for the 3D Meso-NH model to understand the role of biogenic influence on SOA formation in cloud water in tropical environment.

The BIO-MAÏDO project focuses on the effect of cloud on SOA formation in a tropical environment under biogenic influence dominated by isoprene emissions. During recent years, several studies using FT-ICR MS revealed that CHON formulas had high contribution to dissolved organic carbon in cloud water sampling in Colorado, USA (Zhao et al., 2013), NY, USA (Cook et al., 2027), the center of France (Bianco et al., 2018; 2019) and Southern China (Sun et al., 2021; Guo et al., 2023). Similar analysis is underway on three cloud samplings of the BIO-MAÏDO campaign. CHON compounds can have precursors from

biomass burning (BB), anthropogenic and biogenic emissions. For instance, Paglione et al. (2020) observed SOA formation by aqueous phase processing of wood combustion during wintertime in the Po Valley area under influence of fog and low-level clouds. Moreover, part of these compounds present in cloud water can lead to the formation of potentially toxic and harmful aqueous SOA as shown by Witkowski et al. (2022) in the lab, who studied the aqueous oxidation by OH of



nitrophenols. These recent developments in the analysis of the composition of cloud water and the formation of SOA through
aqueous phase chemistry show a need to include more complete chemical mechanisms to understand the role of anthropogenic
and BB influence on SOA formation in cloud/fog water, which is still not well understood. This can be investigated using
available kinetics data and structure activity relationships for such chemical development (Hoffmann et al., 2018; Gonzales-
Sanchez et al., 2021; Li et al., 2023). Future projects involving field campaigns following the BIO-MAÏDO methodology
should be developed to assess a more complete understanding of the influence of cloud chemistry on the formation of AOS.

## Data availability

The data from the BIO-MAÏDO campaign is freely available from https://bio-maido.aeris-data.fr/catalogue/ (last access: 27
July 2023). The 3D simulations were produced with the Meso-NH code version 5.5.0 available at http://mesonh.aero.obs-
mip.fr/ (last access: 20 June 2023).

## Author's contribution

ML is the principal investigator of the BIO-MAÏDO program, who designed the field campaign and prepared the manuscript
with contributions from all the authors. PT organized the field campaign, produced, and organized the strategy for the Meso-
NH simulations, and took part in the analysis of the results shown in the paper. LD led the Task 3 of the project on cloud
sampling and characterization, wrote and reviewed the paper and took part of the field campaign. FB co-led the Task 1
dedicated to atmospheric dynamic and cloud properties, led the tethered balloon operations and the cloud microphysics
instrumentation network. AC was responsible for the VOC measurements, wrote, and reviewed the paper related to gaseous
atmospheric composition analysis and took part of the field experiments. AB co-led the Task 2 of the project related to the
analysis of physico-chemical processes, wrote, and reviewed the paper related to gaseous atmospheric composition analysis
and took part of the field experiments. CJ co-led the Task 2 of the project related to the analysis of physico-chemical processes,
led the HM-3 station, participated to the field experiment and to the data treatment and analysis for this site. VD organized the
field campaign and analyzed the lidar measurements. SH, JLJ and PD supervised the analyses of the off-line PM samples,
processed these data, and commented the overall manuscript. MV co-organized and participated to the cloud droplets sampling
and was responsible of the culture of bacteria from cloud samplings on site. PD supervised the analyses of the off-line PM
samples and some of the analysis of cloud samples, processed these data, and commented the overall manuscript. MR wrote,
and reviewed the paper related to gaseous atmospheric composition analysis and took part of the field experiments. CMV was
involved the analysis of cloud chemistry measurements. SEG was involved in the dynamical analysis of the campaign. MB
participated to the cloud sampling during the field campaign. MF and NM participated to the tethered balloon operations. BV
operated the FLEXPART-AROME model, this included the design of the automated back-trajectory forecasting system which
was operational during the BIO-MAÏDO campaign and providing updated footprints afterwards. CA, NS and BV were



responsible for the deployment, operation, and VOC database generation of the BIRA-IASB PTR-MS located at MO-5. VG
was responsible of the PTR-MS installed in the Atmo-Réunion truck and took part to the field campaign. JMP and MRi
participated to collect the cloud droplets and VOC and OVOC at PO-4. EP participated to the HM-3 station installation and to
the field experiment. EL participated to the field experiment and the HM-3 station disassembly. TB led in situ aerosol
measurements at DOS-2. AR, EM and JB participated to the tethered balloon operations. JMM was in charge of the technical
support at MO-5 for instrumentation deployed for the BIO-MÄIDO campaign. GP, CG, CB, JMT and AT were responsible
for the Atmo-Réunion truck deployment. EF, JLJ and PD wrote and reviewed the paper related to aerosols and clouds
composition analysis. KS was involved in the analysis of aerosol measurements. AMD was involved in the analysis of bacteria
measurements. PA and MJ isolated and identified microbial strains and performed metabarcoding sequencing. JLB performed
the back-trajectories and dynamical analysis. PR, Abi, LD and PD analyzed the cloud samples at the lab. AR was involved in
the dynamical analysis of the campaign. GP was involved in the lidar measurements and their analysis. All reviewed the paper.

**Acknowledgments**

The authors thank AERIS (French national pole for atmospheric services and data: https://www.aeris-data.fr/, last access: 20
June 2023) for his support on data storage. The French Meteorological Office (DIROI/Météo-France) also helped the
management of the campaign by providing a day-by-day meteorological forecasting. The authors gratefully acknowledge
CNRS-INSU (Institut National des Sciences de l'Univers) for supporting measurements performed at MO-5, which is part of
the SI-OPAR (Observatoire de Physique de l'Atmosphère à La Réunion), and those within the long-term monitoring aerosol
program SNO-CLAP (Climate relevant Aerosol Properties from near surface observations), both of which are components of
the ACTRIS French Research Infrastructure and whose data is hosted at the AERIS data center. The authors thank the staff of
UAR3365 in charge of reception at MO-5. Meso-NH simulations have been made on Météo-France supercomputer. Map data
used on Figure 1 are from OpenStreetMap (https://www.openstreetmap.org/copyright/en). Saint-Paul City Hall is thanked for
their support and their authorization to install scientific instrumentation on HM-3 site. All participants in the campaign wish
to thank Doudou for his hospitality on DOS-2 site.

**Financial support**

The BIO-MAÏDO project was funded by the Agence Nationale de la Recherche (ANR-18-CE01-0013). The deployment of
the BIRA-IASB PTR-MS at the Maïdo observatory was supported by the Belgian Federal Science Policy Office (grant no.
BR/175/A2/OCTAVE) with additional funding from Horizon 2020 (grant no. ACTRIS-2 (654109)).



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
