# Peer review of "Measurement Report: Bio-physicochemistry of tropical clouds at Maïdo (Réunion Island, Indian Ocean): overview of results from the BIO-MAÏDO campaign"

_EGUsphere, 2023_

## Referee Comment (RC1)

General comments:

This study by Leriche et al investigated the gas-phase mixing ratio of VOCs and the physical, chemical, and biological properties of aerosols and cloud water in the tropical Réunion areas. Additionally, the authors also presented many auxiliary data, including the turbulent parameters of the boundary layer, radiative fluxes, and emission fluxes of BVOCs from the surrounding vegetation. However, although the manuscript presents some potentially valuable field data, these complex data have not been effectively explained and the manuscript is filled with a large number of inferential expressions. In particular, the lack of necessary connections in the data measured at five different observation sites may confuse readers. Generally, this work may present important and reference field data for future research, and thus is likely of interest to the readership of Atmospheric Chemistry and Physics. I recommend this manuscript for publication after major revisions.

Specific comments.

1. Abstract section.

Line 45: For volatile organic compounds (VOC), the abbreviation of VOCs seems more common.

Line 50: …These air masses likely encountered cloud processing during transport along the slope… How was this inference proposed? as there is no necessary connection between the preceding and following text.

…Chemical composition of particles during the daytime shows a higher concentration of oxalic acid and a more oxidized organic aerosol at MO than at other sites along the slope… What is the indicative significance of this? In addition, what is the situation during the nighttime?

…Despite an in-depth analysis of organic compounds in cloud water, around 80% on average of dissolved organic compounds is undefined highlighting the complexity of the cloud organic matter… This does not seem to be an important conclusion, as the determination of organic matter in cloud water using ultra-high resolution mass spectrometry will inevitably reflect the complexity of cloud water organic matter. I suggest that it is necessary to directly present the molecular composition characteristics of organic compounds.

In particular, it seems better to have a corresponding line number for each line.

2. Introduction section.
Lines 105-110: The BIO-MAÏDO project has been overly described, and in contrast,

the relevant content of this study (line 115) has been oversimplified.

3. Figure 1. I recommend the author to briefly summarize the characteristics of figure 1-5 in the figure caption.

4. Main result section.
It seems that five observation sites showed different data types. Please clarify the connection between the data measured at these observation sites or whether these data can be combined to illustrate several scientific issues?

5. 3.4 Aerosol measurement section.
The authors presented the chemical composition results of PM1 and PM10, but there was more discussion about the chemical compositions in PM10. I am very confused about these discussions. May I ask what is the connection between the chemical composition results of PM1 and PM10, and what atmospheric chemistry issues can be explained?

Lines 465-470: For positive mass factorization (PMF), … Three factors were resolved using PMF analysis…
How did the model run? The author needs to provide at least a detailed explanation of this in the supplementary information.

Lines 485, 495 and 500:…These differences could be related to the…the average concentrations of the sites…
…suggesting that environmental conditions (such as temperature and humidity) can have a role in the emission processes of these compounds by natural sources (e.g., soils, bioaerosols, plants and fungal spores)…
…However, levoglucosan concentrations observed in our study are more likely to be due to domestic biomass burning (e.g. cooking emissions) rather than forest fires (not reported in the area during the campaign)…

There is no evidence or reference for these inferences.

Line 505: …Thus, our results show a strong contribution of biogenic sources on PM10 samples such as fungi spores, soils, and microorganisms and to a lesser extent the contribution from biomass burning aerosols…

Based on the previous discussion (lines 490-550), there seems to be insufficient evidence to support this conclusion.

Lines 610-615: …The detailed presentation of FT-ICR MS results will be reported (work in progress)…
The authors did not present detailed mass spectrometry data but concluded that organic matter in cloud water is complex. I think this is very unreasonable. In addition, for the

conclusion that around 80% on average of dissolved organic compounds is undefined highlighting the complexity of the cloud organic matter. FT-ICR MS analysis about cloud water or rainwater water soluble organic matter has been widely reported. Thus, the fact that organic matter in cloud water is complex does not seem to be an important conclusion of this manuscript in the absence of specific data.

---

## Author Comment (AC1)

**Referee: 1**

Comments to the Author

General comments:

This study by Leriche et al investigated the gas-phase mixing ratio of VOCs and the physical, chemical, and biological properties of aerosols and cloud water in the tropical Réunion areas. Additionally, the authors also presented many auxiliary data, including the turbulent parameters of the boundary layer, radiative fluxes, and emission fluxes of BVOCs from the surrounding vegetation. However, although the manuscript presents some potentially valuable field data, these complex data have not been effectively explained and the manuscript is filled with a large number of inferential expressions. In particular, the lack of necessary connections in the data measured at five different observation sites may confuse readers. Generally, this work may present important and reference field data for future research, and thus is likely of interest to the readership of Atmospheric Chemistry and Physics. I recommend this manuscript for publication after major revisions.

We would like to thank the reviewer for their time spent to review the paper. We considered all the proposed comments/corrections; our answers are indicated in blue in the following.

Specific comments.

1. Abstract section.
Line 45: For volatile organic compounds (VOC), the abbreviation of VOCs seems more common.
As suggested, the abbreviation "VOCs" is now used.

Line 50: ...These air masses likely encountered cloud processing during transport along the slope... How was this inference proposed? as there is no necessary connection between the preceding and following text.
This sentence has been modified as: "These air masses arriving at MO during daytime likely encountered cloud processing during transport along the slope."

...Chemical composition of particles during the daytime shows a higher concentration of oxalic acid and a more oxidized organic aerosol at MO than at other sites along the slope... What is the indicative significance of this? In addition, what is the situation during the nighttime?
The significance of this is that, as oxalic acid is a tracer of cloud processing, its high level at MO can indicate a cloud processing of aerosols sampled at MO. Moreover, observations showed a more oxidized organic aerosol at MO, which is a signature of photochemical aging along the slope. The first sentence has been modified as: "Chemical composition of particles during the daytime shows a higher concentration of oxalic acid, a known tracer of cloud processing, and…"
Night-time composition of PM10 at MO observed during BIO-MAÏDO campaign reflects the composition of free troposphere as expected. We don't think that this is an important result to report in the abstract.

...Despite an in-depth analysis of organic compounds in cloud water, around 80% on average of dissolved organic compounds is undefined highlighting the complexity of the cloud organic matter... This does not seem to be an important conclusion, as the determination of organic matter in cloud water using ultra-high resolution mass spectrometry will inevitably reflect the complexity of cloud water

organic matter. I suggest that it is necessary to directly present the molecular composition characteristics of organic compounds.

We agree with the reviewer that high resolution mass spectrometry will confirm the complexity of cloud water organic matter. Nevertheless, this technique provides molecular formula and elemental ratios, which are useful to explore the variety of the organic matter. In particular, the van Krevelen diagram and the Rivas-Ubach classification, based on the H/C, O/C and N/C ratios, are important tools to evaluate for example the degree of oxidation of the organic matter, which is important to understand cloud reactivity, as well as its composition linked to the sources (lipids, peptides, sugars, condensed aromatics, …). The information obtained with FT-ICR MS could help to drive the next investigation of organic compounds through target analysis of specific molecules of interest. A paper has been recently submitted to EGUsphere (Pailler et al., 2023) where deep analysis of 3 cloud water samples collected during the BIO-MAÏDO campaign was conducted using FT-ICR MS.

We propose to replace this sentence by: "The in-depth analysis of organic compounds in cloud water allowed to characterize around 20% on average of the dissolved organic compounds; additional analysis by ultra-high resolution mass spectrometry will allow to explore the complexity of the missing cloud organic matter."

In particular, it seems better to have a corresponding line number for each line.
The format of line numbering is part of the Word model provided by ACP.

2. Introduction section.
Lines 105-110: The BIO-MAÏDO project has been overly described, and in contrast, the relevant content of this study (line 115) has been oversimplified.

We agree with the reviewer and rewrite the last paragraph of the introduction as follows:

"The aim of the present paper is to present an overview of the results obtained from the campaign including new analyses compared to previous studies (Rocco et al., 2022; Dominutti et al., 2022a; 2022b). First, the general strategy of the campaign and the description of the five sampling sites are provided. Then, main results obtained from measurements are summarized. The main meteorological circulations at synoptic and local scales are analyzed and the dynamical context of a typical cloudy day during the campaign is detailed. Observations of VOCs during the campaign are summarized and compared to other VOCs measurements in Réunion Island. For aerosol observations, after summarizing main results obtained previously, we discuss the comparison between chemical composition from filters sampled at the lowest site and at the Maïdo observatory. Cloud chemistry analyses are summarized, and ongoing work are presented. For biological measurements, we briefly present the diversity of culturable bacteria from cloud samples and the bacteria diversity profiling from aerosol filters. Finally, we discuss in a general perspective obtained results of the campaign."

3. Figure 1. I recommend the author to briefly summarize the characteristics of figure 1-5 in the figure caption.

Figure caption of figure1-5 has been complemented:

"Figure 1. Location of the five-instrumented sites during the BIO-MAÏDO campaign. The grey arrow indicates one of the two main paths for the dominant wind direction. Each site is illustrated with a photography."

"Figure 2. Wind rose at PF-1 and MO-5 between 14 UTC and 04 UTC (nighttime at the top) and from 04 UTC to 18 UTC (daytime at the bottom) averaged over the entire campaign from March 11 to April 7. The intensity is in m s$^{-1}$."

"Figure 3. Natural logarithm of the number of back-trajectory points arriving at Petite France (left) and Maïdo (right) from 15 March to 8 April, per scare of 0.01° (around 1km) size. All trajectory points are

taken into account for the calculation of the total column footprint maps (top), but only trajectory points during the mid-day (between 06 UTC to 14UTC) and between the ground and 500 m agl for the near surface footprint maps (bottom)."

"Figure 4. Meso-NH simulation: top: horizontal cross-section at surface for wind direction (color scale) and intensity (vector in m s-1) and liquid water path (mm, in grey) at 02 UTC (a), 08 UTC (b) for March 28; middle: vertical cross-section along the red line superposed to (a,b) showing the liquid water mixing ratio (color scale in g/kg), wind direction and intensity (vector in m s-1), and TKE (green isoline, in m2 s-2) at 02:00 UTC (c), 08:00 UTC (d) for March 28; bottom:. MARLEY backscattered signals at DOS-2 for March 28 (e)."

"Figure 5: Isoprene, terpenes and BTEX mixing ratio (pptv) for the FARCE campaign (March-April 2015, Duflot et al., 2019), OCTAVE-1 campaign (March-April 2018, Rocco et al., 2020), BIO-MAÏDO campaign (March-April 2019, Rocco et al., 2022) and OCTAVE-2 campaign (March-April 2018-2019, Amelynck et al., 2021) at the Maïdo Observatory."

4. Main result section.
It seems that five observation sites showed different data types. Please clarify the connection between the data measured at these observation sites or whether these data can be combined to illustrate several scientific issues?
As mentioned at the last paragraph of the section 3.1, Rocco et al. (2022) showed a high dynamical connection occurrence between PF-1 and MO-5 during daytime.
Moreover, same data types are observed at several sites. For instance, VOCs mixing ratios were sampled at PF-1, HM-3 and MO-5 using PTR-MS, the $PM_{10}$ mass chemical composition and biological composition were analyzed on filters sampled using the same high-volume sampler, and the visibility was measured with a Present Weather Detector PWD22 (Vaisala) at DOS-2, HM-3, and MO-5.

5. 3.4 Aerosol measurement section.
The authors presented the chemical composition results of PM1 and PM10, but there was more discussion about the chemical compositions in PM10. I am very confused about these discussions. May I ask what is the connection between the chemical composition results of PM1 and PM10, and what atmospheric chemistry issues can be explained?
The details about the NR-PM1 measurements and the analysis of the chemical composition of this data at MO-5 station were thoroughly discussed in a previous paper (Dominutti et al., 2022b).
The PM10 filter measurements were made at the two different sites, MO-5 and PF-1, and this comparison is made in the paper, and therefore much more discussion is included. It is not always evident to compare the two methods as the online PM1 only measures non-refractory species, whereas the PM10 measure both refractory and non-refractory species. However, the evaluation of these two datasets brings complementary insights into the chemical assessment of air masses, the sources, and the processing of atmospheric aerosols.
In the discussion on PM1 (Dominutti et al. 2022b), we focused on the evolution of aerosol chemistry over time based on the different air masses. We were able to put additional emphasis on the fact that there was a higher contribution of more oxidized OA measured in PM1 (or higher f44 fractions) in the latter parts of the field campaign in agreement with filter measurements, showing an increase in the dicarboxylic acids (malic acid and oxalic acids).

Lines 465-470: For positive mass factorization (PMF), ... Three factors were resolved using PMF analysis...
How did the model run? The author needs to provide at least a detailed explanation of this in the supplementary information.

Thanks for the comment. As previously mentioned, the details about the NR-PM1 and the PMF applied to the NR-PM1 can be found in a previous paper (Dominutti et al., 2022).

Briefly, PMF was performed on the organic mass spectrum from m/z 10 to m/z 150; signals after m/z 150 were excluded from the analysis because of low signal-to-noise ratios. A three-factor solution had a Q/Qexp ratio of 0.86 and characterized up to 80% of the total organic aerosol fraction measured. Factor 1 contributed on average 70.5% to the total resolved PMF solutions, had dominant m/z fragments at m/z 28 (36.3%) and m/z 44 (36.3%), and was therefore interpreted as aged/oxidized organic aerosols (MOOA). Factor 2 contributed 11% on average to the total resolved PMF solutions, correlated with reference mass spectra for IEPOXOA (R = 0.3), and had prevalent peaks at m/z 53 (5%) and m/z 82 (1.19%) compared with the other resolved mass spectra. The temporal evolution of this IEPOXOA factor and biogenic age estimation (MVK+MACR+ISOPOOH/isoprene) were similar, indicating that this factor could be interpreted as a secondary organic aerosol derived from biogenic aerosols. Factor 3 contributed to 18.5% on average of the total resolved PMF solutions and showed a good correlation with palmitic acid (R = 0.82) and oleic acid (R = 0.79), as well as primary marine aliphatic-rich organic aerosol factor. Thus, this factor is identified as primary organic aerosols (POA).

We have added more information in the manuscript as follows:

"The contribution of different organic species to the total organic mass concentration was determined using positive mass factorization (PMF), with the source finder (SoFi) toolkit (Canonaco et al., 2013). PMF aims to solve a matrix equation using a weighted least squares approach, which provides different factor profiles by testing rotational techniques available in the ME-2 engine. Different solutions from 2 to 6 factors were tested, and a final solution of 3 factors was chosen based on optimal Q/Qexp values, physically meaningful reference profiles, and time series. The Three factors solution was resolved using PMF analysis on the entire organic matrix from m/z 1 to 150; a more oxidized organic aerosol (MOOA) (75%), a primary organic aerosol (POA) (18.5%), and an isoprene derived organic aerosol (IEPOX-OA) (11%), as fully described in Dominutti et al., (2022b). This solution characterized up to 80% of the total organic aerosol fraction measured."

Lines 485, 495 and 500:...These differences could be related to the...the average concentrations of the sites...

...suggesting that environmental conditions (such as temperature and humidity) can have a role in the emission processes of these compounds by natural sources (e.g., soils, bioaerosols, plants and fungal spores)...

...However, levoglucosan concentrations observed in our study are more likely to be due to domestic biomass burning (e.g. cooking emissions) rather than forest fires (not reported in the area during the campaign)...

There is no evidence or reference for these inferences.

The differences between the sites are associated with the location of the sites. As mentioned in the manuscript, the MO-5 site is under free tropospheric conditions impacting the concentrations of the particles measured during the night (Baray et al., 2013). On the other hand, the meteorological conditions observed at PF-1 were quite different, with a higher impact of air masses and emissions from the surface. The impact of environmental conditions on the emission of sugars, (well-known tracers of soils, plants and fungal spores (Samake et al, 2019 and references therein) has been addressed in the literature, with an increase of the emission of sugar alcohols under an increase of humidity in tropical forests (Zhang et al., (2010)). Even if this hypothesis was not confirmed in the forests around PF-1, this reference suggests a potential explanation together with the dynamic conditions, of the difference in sugars observed between both sites.

Regarding the hypothesis of the levoglucosan associated with cooking emission or long-range transported air masses, this was corroborated due to the absence of fires for the duration of the field

campaign. Additionally, in the article of Rocco et al. (2022), the absence of gaseous tracers (acetonitrile) from biomass burning emissions was reported.

Line 505: ...Thus, our results show a strong contribution of biogenic sources on PM10 samples such as fungi spores, soils, and microorganisms and to a lesser extent the contribution from biomass burning aerosols...
Based on the previous discussion (lines 490-550), there seems to be insufficient evidence to support this conclusion.
Thanks for this comment. The presence of biogenic and biomass-burning tracers in the PM10 filters is observed and discussed in the manuscript. We have corrected the text in order to avoid confusion:
"Thus, our results show a specific contribution of biogenic sources on PM10 samples such as fungi spores, soils, and microorganisms and, to a lesser extent, the contribution from biomass aerosols."

Lines 610-615: ...The detailed presentation of FT-ICR MS results will be reported (work in progress)...
The authors did not present detailed mass spectrometry data but concluded that organic matter in cloud water is complex. I think this is very unreasonable. In addition, for the conclusion that around 80% on average of dissolved organic compounds is undefined highlighting the complexity of the cloud organic matter. FT-ICR MS analysis about cloud water or rainwater water soluble organic matter has been widely reported. Thus, the fact that organic matter in cloud water is complex does not seem to be an important conclusion of this manuscript in the absence of specific data.
We agree with the reviewer that this section regarding FT-ICR MS analysis of cloud samples collected during the campaign is quite unclear. This is related to the fact that when the overview paper has been submitted to the Atmospheric Chemistry and Physics journal, the analysis of the organic matter by the high-resolution mass spectrometry instrument was not achieved. An article has now been submitted to the same journal, presenting the results of the FTICR-MS analyses, and clearly demonstrating the initial conclusions outlined in the overview article. As a result, the corresponding paragraph has been rewritten to clarify the initial conclusions evoked in the present paper as following:
"The contribution of the organic compounds targeted in this project represents around 20% on average of the DOC concentration, reaching up to 35% for one specific cloud. Recently, cloud organic matter has been assessed by high-resolution mass spectrometry using FT-ICR MS instrument (Cook et al., 2017; Bianco et al., 2018; Sun et al., 2021). Thousands of molecular formulae could be potentially detected in cloud water, and their presence and concentration are often related to the influence of primary sources and atmospheric processing. The analysis by FT-ICR MS requires at least 50 mL of cloud water, which is thereafter pre-concentrated and desalted by solid-phase extraction. Three samples (R8, R9, and R10B) presenting enough volume have been analyzed with this instrument equipped with an electrospray ionization (ESI) source, set in the negative ionization mode. The assigned molecular formula retrieved by high-resolution mass spectra does not enable the attribution of a structural formula. Nevertheless, based on the elemental composition, considering the number of carbon, oxygen, nitrogen, hydrogen, and sulfur atoms, the molecular formula can be classified into compounds of biogenic or anthropogenic origin, such as lipids, carbohydrates, proteins or unsaturated hydrocarbons and condensed aromatics, respectively. This analysis offers the possibility of getting information on the DOC fraction not characterized by previously presented targeted analysis. Results on the DOC characterization of the 3 cloud water samples collected at Reunion Island by FT-ICR MS are reported in a paper recently submitted (Pailler et al., submitted, 2023) and compared to the chemical composition of samples collected at the puy de Dôme station (France). Thousands of molecules have been detected which a significant fraction (50%) composed of reduced compounds belonging to lipids probably linked to primary emissions from vegetation, marine surface and urban areas."

References:

Baray, J. L., Courcoux, Y., Keckhut, P., Portafaix, T., Tulet, P., Cammas, J. P., Hauchecorne, A., Godin Beekmann, S., De Mazière, M., Hermans, C., Desmet, F., Sellegri, K., Colomb, A., Ramonet, M., Sciare, J., Vuillemin, C., Hoareau, C., Dionisi, D., Duflot, V., Vérèmes, H., Porteneuve, J., Gabarrot, F., Gaudo, T., Metzger, J. M., Payen, G., Leclair De Bellevue, J., Barthe, C., Posny, F., Ricaud, P., Abchiche, A. and Delmas, R.: Maïdo observatory: A new high-altitude station facility at Reunion Island (21 S, 55 E) for long-term atmospheric remote sensing and in situ measurements, Atmos. Meas. Tech., 6(10), 2865–2877, https://doi.org/10.5194/amt-6-2865-2013, 2013.

Dominutti, P. A., Chevassus, E., Baray, J.-L., Jaffrezo, J.-L., Borbon, A., Colomb, A. Deguillaume, L., El Gdachi, S., Houdier, S., Leriche, M., Metzger, J.-M., Rocco, M., Tulet, P., Sellegri, K., and Freney, E.: Evaluation of sources, precursors, and processing of aerosols at a high-altitude tropical site, ACS Earth Space Chem., 6, 2412–2431, https://doi.org/10.1021/acsearthspacechem.2c00149, 2022b.

Pailler, L., Deguillaume, L., Lavanant, H., Schmitz, I., Hubert, M., Nicol, E., Ribeiro, M., Pichon, J.-M., Vaïtilingom, M., Dominutti, P., Burnet, F., Tulet, P., Leriche, M., and Bianco, A.: Molecular composition of clouds: a comparison between samples collected at tropical (Réunion Island, France) and mid-north (puy de Dôme, France) latitudes, EGUsphere [preprint], https://doi.org/10.5194/egusphere-2023-2706, 2023.

Rocco, M., Baray, J.-L., Colomb, A., Borbon, A., Dominutti, P., Tulet, P., Amelynck, C., Schoon, N., Verreyken, B., Duflot, V., Gros, V., Sarda-Estève, R., Péris, G., Guadagno, C., and Leriche, M.: High resolution dynamical analysis of Volatile Organic Compounds (VOC) measurements during the BIO-MAÏDO field campaign (Réunion Island, Indian Ocean), J. Geophys. Res. Atmos., 127, e2021JD035570, https://doi.org/10.1029/2021JD035570, 2022.

Samaké, A., Jaffrezo, J.-L., Favez, O., Weber, S., Jacob, V., Albinet, A., Riffault, V., Perdrix, E., Waked, A., Golly, B., Salameh, D., Chevrier, F., Oliveira, D. M., Bonnaire, N., Besombes, J.-L., Martins, J. M. F., Conil, S., Guillaud, G., Mesbah, B., Rocq, B., Robic, P.-Y., Hulin, A., Meur, S. L., Descheemaecker, M., Chretien, E., Marchand, N., and Uzu, G.: Polyols and glucose particulate species as tracers of primary biogenic organic aerosols at 28 French sites, Atmos. Chem. Phys., 19, 3357–3374, https://doi.org/10.5194/acp-19-3357-2019, 2019.

Zhang, T., Engling, G., Chan, C.-Y., Zhang, Y.-N., Zhang, Z.-S., Lin, M., Sang, X.-F., Li, Y. D., and Li, Y.-S.: Contribution of fungal spores to particulate matter in a tropical rainforest, Environ. Res. Lett., 5, 024010, https://doi.org/10.1088/1748-9326/5/2/024010, 2010.

**Referee: 2**

Comments to the Author

General

This is a comprehensive and really interesting measurement report (MR) on investigations of tropical clouds at Maido on the reunion Islands in 2019. See my second comment under 'details' for the approach. I am not sure why this should be a 'measurement report' rather than a regular paper. Is that really the best choice?

Other than this, I congratulate the authors for having undertaken this campaign and making the data available through this MR. I recommend acceptance subject to only minor amended discussions and some structural improvements according to my remarks below.

We would like to thank the reviewer for their time spent to review the paper. The paper was submitted as a measurement report upon suggestion of the editor. We considered all the proposed comments/corrections; our answers are indicated in blue in the following.

Details

Abstract: I feel the abstract is ok for a MR, giving an overview of the measurement results provided.

line 120 ff, section 2, paragraph 2.1: If I look at Figure 1, interestingly, the measurement station there are aligned along the main wind direction, so that possibly processing patters along the sequence of stations from Petite-Franc to the mobile lab to the Domaine des Orchidées sauvages to Hotel de Maido to Piste Omega and, finally, Maido observatory will become observable. However, I cannot find this thought in this early section and I clearly suggest to add this here, early in the manuscript.

Doesn't this natural set-up ask for checking the results on whether multiphase processing of gases and particles can be observed ? Or is this beyond the scope of an MR ? But the authors are motivating their measurements here with the points (1) to (3), so maybe such a thought like 'processing along a Lagrange-typ trajectory' could be added?

We agree with the reviewer and thank them for the suggestion. We modify paragraph beginning at line 134 as:

"The field campaign included five sampling sites (Figure 1). Except for the Maïdo observatory, these sites are all located along the northwestern slope to the Maïdo site, identified as one of the two main paths for dominant winds (Duflot et al., 2019). At the mid-morning almost every day, clouds form on the slope of the Maïdo, and in general evaporate at the altitude of the observatory (Duflot et al., 2019). The location of the sampling sites should allow observing the air mass processing by clouds along the slope in a lagrangian trajectory approach where the Maïdo observatory is the receptor site."

In the section 2.2. to 2.6 the above-mentioned measurement stations or observatories are described in detail with a comprehensive description if the measurements possibilities and their sampling frequencies.

l 304: Are these average wind roses consistent with the arrow in Figure 1 which shows like north-west as the main wind direction ? If the stations are all subject to incoming westerly winds (during daytime), one would expect that they are all measuring the same. Then, a processing sequence would be difficult to obtain. In the time series where data were obtained, is there periods when NW in flow prevailed ? Can you identify these periods ?

We added in the legend of Fig. 1 the information that the grey arrow is one of the two main paths for the dominant wind as observed by Duflot et al. (2019). The periods where the NW wind direction is dominant during daytime correspond to days with the highest dynamic connection occurrence (see your next answer under).

l268 ff, section 3.1: Following my above comment: Is it the results of this section that March 16th and April 1st were clearly the best periods where a sequential passing of the station was identified ? Could you add a bit more in the present paper rather than mainly referring to Rocco et al (2022) ?
It would be great if this argumentation would be accessible in the MR alone. Re that Rocco et al Figure: Wouldn't March 31st be nearly a s good as April 1 ? What is the selection parameter ?

[Figure]

Recently, Manon Rocco submitted to JGR an erratum to Rocco et al. (2022). She is waiting for the editor's answer. Then, the article will be directly corrected online, and an error notice will be added. The error concerned the calculation of forward and backward trajectories in the MESO-CAT model. The updated calculations and trajectories lead to a more accurate interpretation of the link of trajectories and VOCs measurements during the measurement campaign.
Here under is the corrected Figure 4C.

[Figure]

Figure 4C (Rocco et al., 2022): Dynamic connection occurrence (%) from 26 March to 6 April considering both 500 m and 100 m domains with 15-minute time resolution trajectories. Top: Daily variation of dynamic connection occurrence from 0h UT to 23h UT. Bottom: Dynamic connection occurrence for each day.

These updated results are more consistent with the Figure 5 from Rocco et al. (2022). Figure 5 showed at PF a mean wind direction of W-NW in the morning, which became mainly of W in the second part of daytime. The top panel of Figure 4C showed higher dynamic connection during the morning than during the afternoon between PF and MO.

In regard of this correction on Figure 4C of Rocco et al. (2022) the last paragraph of section 3.1 was modified as:

"The connection between MO-5 and the other observation sites of the campaign is clearly evidenced by the footprints. This is consistent with the high dynamic correction occurrence computed by Rocco et al. (2022, Figure 4C) for almost each day of the campaign. As for PF-1, several trajectories also indicate an origin of the marine boundary layer. These specific periods will be studied preferentially to follow the lagrangian evolution of the chemical composition of the air mass."

l 351: It is written that ...this simulation was typical and highlighted the good connection between the observation sites." Is a deeper analysis foreseen here ? Can the connection be proven also by gas phase or particle phase constituents' measurements ?

Figure 4C above shows a high dynamic connection on 28 March 2019 between PF and MO. Supplementary materials of Rocco et al. (2022) had been also corrected and the corrected backward and forward trajectories are reproduced under. We consider that these detailed trajectories available with the Rocco et al.'s paper are sufficient to justify the choice of 28 March 2019.

**Backward – 28/03/2019**

[Figure]

[Figure]

[Figure]

Figure S.3. Forward and backward trajectories from PF and MO in the 500 m spatial resolution domain on 28 March 2019 from 0h UT to 22h UT every 2 hours.

l 389: The VOC measurement contain a wealth of data and the authors are congratulated to have mesured this dataset. What about OVOC ? Are thes a separate dataset ? In line 404 Rocco at al (2020) is referenced which is on HCHO, which is a carbonyl - what about other carbonyl compounds ? Maybe VOCs and OVOC can be treated separately. The OVOC would be very important to follow VOC oxidation and they also link the gas phase oxidations to particle phase compositions.

We agree with the importance of looking at OVOC as oxidation products of primary VOC and precursors of SOA. Methanol and acetone were studied in Rocco et al. (2022) and OVOC were indicated in Figure 6 as the sum of methanol, acetaldehyde, acetone, and methyl ethyl ketone.

Unfortunately, until now, HCHO mixing ratios sampled with an aerolaser during BIO-MAÏDO at Petite-France from 28 March to 4 April 2019 (Table 1) and at Maïdo observatory from 13 to 27 March 2029 (Table 5) were not exploited. Table S1 shows that aerolaser worked correctly on 23 and 24 March at observatory (MO) and on 31 March and on 3 and 4 April 2019 at Petite-France (PF). Figures above show the aerolaser measurements at PF on 31 March and at MO on 23 and 24 March as examples.

These figures seem to show a higher level of HCHO at MO than at PF. The order of magnitude observed at MO are consistent with observations of HCHO mixing ratio during the FARCE campaign (Duflot et al., 2019).

[Figure]

HCHO mixing ratios (ppbv) at Petite-France on 31 March 2019.

[Figure]

HCHO mixing ratios (ppbv) at Maïdo observatory on 23 and 24 March 2019.

l453: 'Aerosol': Maybe here subsections for the physical measurements and chemical aerosol particle characterization can be introduced ?

An introduction was added to this section:

"During the BIO-MAÏDO campaign, an online ToF-ACSM, operating continuously from March 13th to April 2nd, was used to determine the chemical composition of non-refractory- PM1 (NR-PM1) aerosol at MO-5, providing mass concentrations for organic, sulphate, nitrate, ammonium and chloride species. Additionally, PM10 aerosols were simultaneously sampled by offline filters at MO-5 and PF-1 during the whole field campaign."

And the rest of the text was modified accordingly.

l534 ff: The sections of Cloudwater and Biological systems also contain a wealth of data. It would be great to clearly state what has been analyzed and if a deeper analysis is under preparation and for what exactly. This way, the present MR could guide the reader for future evaluations which use the campaign data but which are not in the scope of this present MR.

As mentioned in the text, this section mainly summarized results obtained by Dominutti et al. (2022a). This cloud water analysis has been complemented by FT-ICR MS measurements, which were integrated in a recent manuscript just submitted to ACP (Pailler et al., 20236). Moreover, a 0D modelling study using cloud-water data is in progress as mentioned in the conclusion of the MR.

References:

Duflot, V., Tulet, P., Flores, O., Barthe, C., Colomb, A., Deguillaume, L., Vaïtilingom, M., Perring, A., Huffman, A., Hernandez, M. T., Sellegri, K., Robinson, E., O'Connor, D. J., Gomez, O. M., Burnet, F., Bourrianne, T., Strasberg, D., Rocco, M., Bertram, A. K., Chazette, P., Totems, J., Fournel, J., Stamenoff, P., Metzger, J.-M., Chabasset, M., Rousseau, C., Bourrianne, E., Sancelme, M., Delort, A.-M., Wegener, R. E., Chou, C., and Elizondo, P.: Preliminary results from the FARCE 2015 campaign: multidisciplinary study of the forest–gas–aerosol–cloud system on the tropical island of La Réunion, Atmos. Chem. Phys., 19, 10591–10618, https://doi.org/10.5194/acp-19-10591-2019, 2019.

Pailler, L., Deguillaume, L., Lavanant, H., Schmitz, I., Hubert, M., Nicol, E., Ribeiro, M., Pichon, J.-M., Vaïtilingom, M., Dominutti, P., Burnet, F., Tulet, P., Leriche, M., and Bianco, A.: Molecular composition of clouds: a comparison between samples collected at tropical (Réunion Island, France) and mid-north (puy de Dôme, France) latitudes, EGUsphere [preprint], https://doi.org/10.5194/egusphere-2023-2706, 2023.

Rocco, M., Baray, J.-L., Colomb, A., Borbon, A., Dominutti, P., Tulet, P., Amelynck, C., Schoon, N., Verreyken, B., Duflot, V., Gros, V., Sarda-Estève, R., Péris, G., Guadagno, C., and Leriche, M.: High resolution dynamical analysis of Volatile Organic Compounds (VOC) measurements during the BIO-MAÏDO field campaign (Réunion Island, Indian Ocean), J. Geophys. Res. Atmos., 127, e2021JD035570, https://doi.org/10.1029/2021JD035570, 2022.

---

## Editor Decision (ED1)

Please see further comments directly in the submission form. Additionally, please see tracked changes below to be considered as one strategy to reduce the word count in the abstract. As submitted in the revised version of the manuscript, the abstract contained ~370 words, which is still well above the 250 word limit suggested by the journal. I have made changes below that streamline the text somewhat, without significantly altering the meaning. It now stands at 269 words, and so is only minimally above the suggested limit. Please review and accept the changes below, if you agree that the meaning is not changed. Whether through these or separate changes, please keep the word limit to below or near the 250 word limit https://www.atmospheric-chemistry-and-physics.net/policies/guidelines_for_authors.html

The BIO-MAÏDO (Bio-physicochemistry of tropical clouds at Maïdo : processes and impacts on secondary organic aerosols formation) campaign was conducted from  13th  March to  4th  April 2019 on the tropical Réunion Island . The main objective of the project was to improve  understanding of cloud impacts on the formation of secondary organic aerosols (SOA) from biogenic volatile organic compound (BVOC) precursors in a tropical environment. Instruments were deployed at five sites:  receptor site,  Maïdo observatory (MO) at 2165 m asl and four sites along the slope of the Maïdo mountain.  Observations include measurements  of volatile organic compounds (VOCs) and characterization of the physical, chemical, and biological (bacterial diversity and culture-based approaches) properties of aerosols and cloud water.Turbulent parameters of the boundary layer, radiative fluxes, and emissions fluxes of BVOCs from the surrounding vegetation were measured to help  interpret observed chemical concentrations in the different phases. Dynamical analyses show two preferred trajectories routes for air masses arriving at MO during the daytime. Both trajectories correspond to  return branches of the trade winds associated with  up-slope thermal breezes, where air masses  likely encountered cloud-processing . The highest mixing ratio of oxygenated VOCs (OVOCs) were measured above the site located in the endemic forest and the highest contribution of OVOCs to total VOCs at MO. Chemical composition of particles during  daytime showe higher concentration of oxalic acid, a  tracer of cloud processing and photochemical age, and a more oxidized organic aerosol at MO than at other sites along Approximately 20%  of the dissolved organic compounds were analyzed. Additional analyse by ultra-high resolution mass spectrometry will  explore the complexity of the missing cloud organic matter.

---

## Author Response (AR2)

We would like to thank the editor for their time spent to revise the abstract. The new abstract almost identical to the editor's suggested text is provided below:

"The BIO-MAÏDO (Bio-physicochemistry of tropical clouds at Maïdo: processes and impacts on secondary organic aerosols formation) campaign was conducted from 13 March to 4 April 2019 on the tropical Réunion Island. The main objective of the project was to improve understanding of cloud impacts on the formation of secondary organic aerosols (SOA) from biogenic volatile organic compounds (BVOCs) precursors in a tropical environment. Instruments were deployed at five sites: receptor site, Maïdo observatory (MO) at 2165 m asl and four sites along the slope of the Maïdo mountain. Observations include measurements of volatile organic compounds (VOCs) and characterization of the physical, chemical, and biological (bacterial diversity and culture-based approaches) properties of aerosols and cloud water. Turbulent parameters of the boundary layer, radiative fluxes, and emissions fluxes of BVOCs from the surrounding vegetation were measured to help interpret observed chemical concentrations in the different phases. Dynamical analyses showed two preferred trajectories routes for air masses arriving at MO during the daytime. Both trajectories correspond to return branches of the trade winds associated with up-slope thermal breezes, where air masses likely encountered cloud processing. The highest mixing ratio of oxygenated VOCs (OVOCs) were measured above the site located in the endemic forest and the highest contribution of OVOCs to total VOCs at MO. Chemical composition of particles during daytime showed higher concentrations of oxalic acid, a tracer of cloud processing and photochemical aging, and a more oxidized organic aerosol at MO than at other sites. Approximately 20% of the dissolved organic compounds were analyzed. Additional analyses by ultra-high resolution mass spectrometry will explore the complexity of the missing cloud organic matter."